# EXACT: Scalable Graph Neural Networks Training via Extreme Activation Compression

**Zirui Liu[1], Kaixiong Zhou[1], Fan Yang[1], Li Li[2], Rui Chen[2*], and Xia Hu[1*]**
[1]Rice University, [2]Samsung Research America
`{zl105, kaixiong.zhou, fy19, xia.hu}@rice.edu,`
`{li.li1, rui.chen1}@samsung.com`

## ABSTRACT

Training Graph Neural Networks (GNNs) on large graphs is a fundamental challenge due to the high memory usage, which is mainly occupied by activations (e.g., node embeddings). Previous works usually focus on reducing the number of nodes retained in memory. In parallel, unlike what has been developed for other types of neural networks, training with compressed activation maps is less explored for GNNs. This extension is notoriously difficult to implement due to the lack of necessary tools in common graph learning packages. To unleash the potential of this direction, we provide an optimized GPU implementation which supports training GNNs with compressed activations. Based on the implementation, we propose a memory-efficient framework called "EXACT", which for the first time demonstrates the potential and evaluates the feasibility of training GNNs with compressed activations. We systematically analyze the trade-off among the memory saving, time overhead, and accuracy drop. In practice, EXACT can reduce the memory footprint of activations by up to $32\times$ with 0.2-0.5% accuracy drop and 10-25% time overhead across different models and datasets. We implement EXACT as an extension for Pytorch Geometric and Pytorch. In practice, for Pytorch Geometric, EXACT can trim down the hardware requirement of training a three-layer full-batch GraphSAGE on *ogbn-products* from a 48GB GPU to a 12GB GPU. The code is available at `https://github.com/warai-0toko/Exact`.

## 1 INTRODUCTION

Despite Graph Neural Networks (GNNs) have achieved great success across different graph-related tasks, training GNNs on large graphs is a long-standing challenge due to its extensive memory requirement (Kipf & Welling, 2017; Zhang & Chen, 2018; Cai et al., 2021b). The extensive memory consumption of a GNN stems from its recursive neighborhood aggregation scheme, where each node aggregates the embeddings of its neighbors to update its new embedding at each layer. Thus, training an $L$-layer GNN requires storing all $L$ layers' intermediate node embeddings in GPU memory for computing the gradients, and this typically adds several times more memory than holding the node feature matrix (see Algorithm 1 for a detailed analysis). Hence, storing these node embeddings is the major memory bottleneck for training GNNs on large graphs.

Most of the existing works towards this problem can be roughly divided into two categories. First, some works propose to train GNNs with sampled subgraphs instead of the whole graph at each step. In this way, only node embeddings that are present in the current subgraph will be retained in memory (Chiang et al., 2019; Hamilton et al., 2017; Zeng et al., 2020; Zou et al., 2019; Chen et al., 2018; Huang et al., 2018). Second, another line of works tries to decouple the neighborhood aggregation from prediction, either as a preprocessing step (Wu et al., 2019; Klicpera et al., 2018; Yu et al., 2020) or post-processing step (Huang et al., 2020), where the model is simplified as the Multi-Layer Perceptron (MLP) that can be trained with mini-batch data.

---

*Corresponding Author

In parallel, another orthogonal direction is to store only the compressed node embeddings (i.e., activations) in memory for computing the gradients. Recent works propose to quantize the activations in lower numerical precision (e.g., using 8-bit integer) during the forward pass (Chakrabarti & Moseley, 2019; Fu et al., 2020; Chen et al., 2021a; Evans & Aamodt, 2021). This framework successfully trims down the memory requirement for training Convolutional Neural Networks (CNNs) by a large margin, at the cost of additional time overhead and loss of accuracy. Ideally, the real-world usage requires that the training method should achieve a balanced trade-off among the following three aspects: **1. Space.** It should enable to train GNNs on large graphs using off-the-shelf hardwares, such as GPUs and CPUs; **2. Speed.** The time overhead should be acceptable, ideally as small as possible; **3. Model Performance.** The loss of accuracy should be acceptable, ideally as small as possible.

Although storing the compressed activations successfully saves the memory for CNNs, up to our knowledge, there is no existing work that extends this direction to GNNs and evaluates the mentioned trade-off to analyze its feasibility. Despite the extension is conceptually straightforward, this direction is less-explored since it can be notoriously difficult to implement to fully leverage hardware potentials. This dilemma stems from the fact that the necessary tools for supporting this direction are usually missing in common graph learning packages. For example, operations in popular graph learning packages only support casting tensors down to 8-bit integer on GPUs, significantly limiting the memory saving potential (Paszke et al., 2019; Fey & Lenssen, 2019; Wang et al., 2019). As a result, previous GNN quantization works either emulate inference-time quantization via "simulated quantization" (Tailor et al., 2021; Zhao et al., 2020), or are impractical to use GPUs for accelerating (Feng et al., 2020). To unleash the potential of this direction, we provide a space-efficient GPU implementation for supporting common operations in GNNs with compressed activations. Equipped with our implementation, this paper asks: *To what extent can we compress the activations with both acceptable loss in accuracy and time overhead for scalable GNN training?*

To answer the open question, we first explore two different types of compression methods. One is "quantization" that compresses the activations into lower numerical precision. The other one is called "random projection" (Achlioptas, 2001) that projects the activations into a low-dimensional space. Both these two simple strategies can achieve near-lossless accuracy at a non-trivial compression ratio. For example, the loss in accuracy is negligible ($0.2\%$) even under the vanilla 2-bit quantization. However, we cannot further push forward the memory saving by these two methods, e.g., we cannot use a numerical precision below 1-bit. Considering that the real-world graphs often contain hundreds of millions of nodes, **our main goal** is to trim down the memory consumption to the maximum extent among the three aspects, as long as the other two are acceptable. We then naturally explore the direction of combining random projection and quantization, dubbed "EXACT", to aggressively maximize the memory saving. EXACT essentially applies random projection and quantization sequentially for compressing activations. Despite the superior memory saving, another following question is *whether the combination brings significantly worse model performance and larger time overhead?* Following the questions, we make three major contributions as follows:

- We provide a space-efficient GPU implementation for training GNNs with compressed activations as an extension for Pytorch. Based on our implementation, we are the first one training GNNs with compressed activations and demonstrating its potential for real-world usage. EXACT can complement the existing studies, as it can be integrated with most of existing solutions.

- We propose EXACT, a simple-yet-effective framework which applies random projection and quantization sequentially on activations for scalable GNN training. We theoretically and experimentally show that applying random projection and quantization sequentially has an "interaction" effect. Namely, from the model performance aspect, after random projection, applying quantization only has a limited impact on the model performance. From the time aspect, EXACT runs comparably or even faster than quantization only.

- Despite the simplicity, EXACT achieves non-trivial memory saving with both acceptable time overhead and loss in accuracy: EXACT can reduce the memory footprint of activations by up to $32\times$ with roughly $0.5\%$ loss in accuracy and $10 - 25\%$ time overhead across models and datasets. We implement EXACT as an extension for Pytorch Geometric and Pytorch. For Pytorch Geometric, EXACT trims down the hardware requirement of training a full-batch GraphSAGE (Hamilton et al., 2017) on *ogbn-products* (Hu et al., 2020) from a 48GB GPU to a 12GB GPU.

## 2 THE MEMORY CONSUMPTION OF GNNS

**Background.** Let $\mathcal{G} = (\mathcal{V}, \mathcal{E})$ be an undirected graph with $\mathcal{V} = (v_1, \cdots, v_{|\mathcal{V}|})$ and $\mathcal{E} = (e_1, \cdots, e_{|\mathcal{E}|})$ being the set of nodes and edges, respectively. Let $\boldsymbol{X} \in \mathbb{R}^{|\mathcal{V}| \times d}$ be the node feature matrix of the whole graph. The graph structure can be represented by an adjacency matrix $\boldsymbol{A} \in \mathbb{R}^{|\mathcal{V}| \times |\mathcal{V}|}$, where $\boldsymbol{A}_{i,j} = 1$ if $(v_i, v_j) \in \mathcal{E}$ else $\boldsymbol{A}_{i,j} = 0$. In this work, we are mostly interested in the task of *node classification*, where the goal is to learn the representation $\boldsymbol{h}_v$ for all $v \in \mathcal{V}$ such that the label $y_v$ can be easily predicted. To obtain such a representation, GNNs follow the neighborhood aggregation scheme. Specifically, GNNs recursively update the representation of a node by aggregating the representations of its neighbors. Formally, the $l^{\text{th}}$ layer of a GNN can be written as $\boldsymbol{h}_v^{(l+1)} = \text{UPDATE}\left(\boldsymbol{h}_v^{(l)}, \bigoplus_{u \in \mathcal{N}(v)} \text{MSG}\left(\boldsymbol{h}_u^{(l)}, \boldsymbol{h}_v^{(l)}\right)\right)$, where $\boldsymbol{h}_v^{(l)}$ is the representation of node $v$ at the $l^{\text{th}}$ layer. $\mathcal{N}(v)$ denotes the neighboring nodes of node $v$, not including $v$ itself. The full table of notations can be found in Appendix A Table 5. For node $v$, messages from its neighbors are calculated by the message function $\text{MSG}(\cdot)$. Then these messages are aggregated using a permutation-invariant aggregation function $\bigoplus$. The aggregated features at $v$ are updated by $\text{UPDATE}(\cdot)$. For example, the Graph Convolutional Network (GCN) (Kipf & Welling, 2017) layer can be defined as

$$\boldsymbol{H}^{(l+1)} = \text{ReLU}(\hat{\boldsymbol{A}} \boldsymbol{H}^{(l)} \boldsymbol{\Theta}^{(l)}), \tag{1}$$

where $\boldsymbol{H}^{(l)}$ is the node embedding matrix consisting of all nodes' embeddings at the $l^{\text{th}}$ layer and $\boldsymbol{H}^{(0)} = \boldsymbol{X}$. $\boldsymbol{\Theta}^{(l)}$ is the weight matrix of the $l^{\text{th}}$ GCN layer. $\hat{\boldsymbol{A}} = \tilde{\boldsymbol{D}}^{-\frac{1}{2}} \boldsymbol{A} \tilde{\boldsymbol{D}}^{-\frac{1}{2}}$ is the normalized adjacency matrix, where $\tilde{\boldsymbol{D}}$ is the degree matrix of $\boldsymbol{A} + \boldsymbol{I}$. We note that $\hat{\boldsymbol{A}}$ is usually stored using the sparse matrix format. For GCN layers, the message and the aggregation function are fused into a Sparse-Dense Matrix Multiplication (SPMM [1]) operation. Namely, $\hat{\boldsymbol{A}} \boldsymbol{H}^{(l)} \boldsymbol{\Theta}^{(l)} = \text{SPMM}(\hat{\boldsymbol{A}}, \boldsymbol{H}^{(l)} \boldsymbol{\Theta}^{(l)})$. Thus, the computation graph of Equation 1 can be written as

$$\boldsymbol{H}^{(l+1)} = \text{ReLU}\left(\text{SPMM}\left(\hat{\boldsymbol{A}}, \text{MM}(\boldsymbol{H}^{(l)}, \boldsymbol{\Theta}^{(l)})\right)\right), \tag{2}$$

where $\text{MM}(\cdot, \cdot)$ is the normal Dense-Dense Matrix Multiplication. Equation 2 resembles how GCNs are implemented in popular packages (Fey & Lenssen, 2019; Wang et al., 2019).

Here we analyze the memory consumption for the forward pass of GCN since most of the memory is occupied during the forward pass. For the memory usage of the backward pass, a detailed analysis is given in Appendix C. Specifically, for an $L$ layer GCN, suppose the hidden dimensions of layer $0, \cdots, L-1$ are the same, which are denoted as $D$. The forward pass of GCN layers is shown in Appendix C Algorithm 1. For layer $l$, it saves the following four variables in memory. **(1)** the weight matrix $\boldsymbol{\Theta}^{(l)} \in \mathbb{R}^{D \times D}$ whose shape is independent to the graph size and is generally negligible. **(2)** the normalized adjacency matrix $\hat{\boldsymbol{A}}$ (in CSR format) whose space complexity is $\mathcal{O}(|\mathcal{V}| + |\mathcal{E}|)$. Note that it only needs to store one $\hat{\boldsymbol{A}}$ in memory which can be accessed by different layers. Thus, the memory consumption of $\hat{\boldsymbol{A}}$ is independent of the number of layers and is not the main memory bottleneck. **(3)** the intermediate result $\boldsymbol{J}^{(l)} = \text{MM}(\boldsymbol{H}^{(l)}, \boldsymbol{\Theta}^{(l)}) \in \mathbb{R}^{|\mathcal{V}| \times D}$. For an $L$ layer GCN, storing $\{\boldsymbol{J}^{(0)}, \cdots \boldsymbol{J}^{(L-1)}\}$ has a $\mathcal{O}(L|\mathcal{V}|D)$ space complexity, which is the main memory bottleneck. **(4)** the node embedding matrix $\boldsymbol{H}^{(l)} \in \mathbb{R}^{|\mathcal{V}| \times D}$. For an $L$ layer GCN, storing $\{\boldsymbol{H}^{(0)}, \cdots \boldsymbol{H}^{(L-1)}\}$ has a $\mathcal{O}(L|\mathcal{V}|D)$ space complexity, which is also a main memory bottleneck. In this paper, we use the term "activation maps" to encompass $\boldsymbol{H}$, $\boldsymbol{J}$, and activation maps of other commonly used layers/operations such as BatchNorm, ReLU and Dropout.

## 3 METHODOLOGY

From the above analysis, the memory consumption of GCNs can be significantly reduced by storing only the compressed activations. As shown in Figure 1, due to the simplicity and small time overhead of quantization (Section 3.1) and random projection (Section 3.2), we first explore these two methods for compressing the activations of GNNs. We show that these two simple methods can

---

[1] We note that the output of $\text{SPMM}(\cdot, \cdot)$ is a dense matrix.

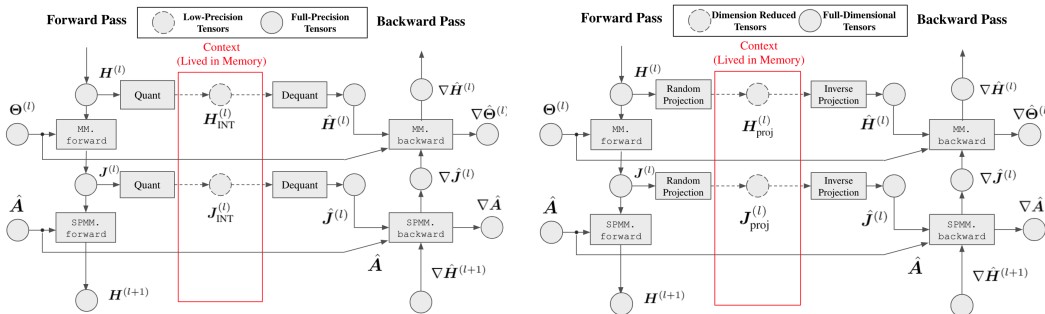

(a) GCN layers with quantized activations.          (b) GCN layers with randomly projected activations.

Figure 1: The training procedure of GCN layers, where only compressed activations are stored in memory. For illustration convenience, ReLU is ignored. During the forward pass, each layer's *accurate activations* ($\boldsymbol{H}^{(l)}$) is used to compute those for the subsequent layer. Then the accurate activations will be compressed into the *compressed activations* ($\boldsymbol{H}_{\text{INT}}^{(l)}$ and $\boldsymbol{H}_{\text{proj}}^{(l)}$), which overwrites the the accurate activations and is retained in the memory. During the backward pass, we recover the compressed activations back to *decompressed activation* ($\hat{\boldsymbol{H}}^{(l)}$) for computing the gradient.

bring moderate compression ratios with negligible loss in accuracy. However, the highest compression rates of these two methods are limited by either hardwares or the accuracy drop. Motivated by GNNs' endless demand on memory, we then explore the direction of combining quantization with random projection into one framework termed "EXACT" to aggressively maximize the memory saving (Section 3.3). We show that random projection and quantization have an "interaction" effect. That is, from the model performance aspect, after random projection, applying quantization only has a limited impact on the model performance. From the time aspect, EXACT runs comparably or even faster than quantization only.

## 3.1 STORING QUANTIZED ACTIVATIONS

Inspired by ActNN (Chen et al., 2021a), as shown in Figure 1a, we first explore the direction of compressing activation maps in lower numerical precision. Specifically, during the forward pass, each layer's *accurate activations* (e.g., $\boldsymbol{J}^{(l)}$ and $\boldsymbol{H}^{(l)}$) is used to compute those for the subsequent layer. Then the accurate activations will be quantized into the *compressed activations* (e.g., $\boldsymbol{H}_{\text{INT}}^{(l)}$ and $\boldsymbol{J}_{\text{INT}}^{(l)}$) with lower numerical precision, which overwrite the accurate activations and are retained in the memory. During the backward pass, we dequantize the compressed activations back to full-precision (e.g., $\hat{\boldsymbol{H}}^{(l)}$ and $\hat{\boldsymbol{J}}^{(l)}$). Then the gradients are computed based on the dequantized activations. We note that all operations (e.g., MM and SPMM) are done in full-precision since most of the GPUs do not support operands with bit-width other than full-precision and half-precision. For convenience, we use the term **"precision" to present "numerical precision"** in the following of this paper. Below we introduce the technical details of (de)quantization.

Specifically, each node embedding vector $\boldsymbol{h}_v^{(l)}$ will be quantized and stored using $b$-bit integers. Let $B = 2^b - 1$ be the number of quantization bins. The integer quantization can be expressed as

$$\boldsymbol{h}_{v_{\text{INT}}}^{(l)} = \text{Quant}(\boldsymbol{h}_v^{(l)}) = \lfloor \frac{\boldsymbol{h}_v^{(l)} - Z_v^{(l)}}{r_v^{(l)}} B \rceil, \tag{3}$$

where $Z_v^{(l)} = \min\{\boldsymbol{h}_v^{(l)}\}$ is the zero-point, $r_v^{(l)} = \max\{\boldsymbol{h}_v^{(l)}\} - \min\{\boldsymbol{h}_v^{(l)}\}$ is the range for $\boldsymbol{h}_v^{(l)}$, and $\lfloor \cdot \rceil$ is the stochastic rounding operation (Courbariaux et al., 2015). $\boldsymbol{H}_{\text{INT}}^{(l)}$ in Figure 1a is the matrix containing all quantized $\boldsymbol{h}_{v_{\text{INT}}}^{(l)}$. During the backward pass, each $\boldsymbol{h}_{v_{\text{INT}}}^{(l)}$ will be dequantized as

$$\hat{\boldsymbol{h}}_v^{(l)} = \text{Dequant}(\boldsymbol{h}_{v_{\text{INT}}}^{(l)}) = r_v^{(l)} \boldsymbol{h}_{v_{\text{INT}}}^{(l)}/B + Z_v^{(l)}. \tag{4}$$

$\hat{\boldsymbol{H}}^{(l)}$ in Figure 1a is the matrix containing all dequantized embeddings $\hat{\boldsymbol{h}}_v^{(l)}$. The following proposition characterizes the effect of quantization, which is adopted from Chen et al. (2021a).

**Proposition 1 (Details in Appendix E)** *The above quantization and dequantization are unbiased operations, i.e.,* $\mathbb{E}[\hat{\boldsymbol{h}}_v^{(l)}] = \mathbb{E}[\text{Dequant}(\text{Quant}(\boldsymbol{h}_v^{(l)}))] = \boldsymbol{h}_v^{(l)}$ *and* $\text{Var}(\hat{\boldsymbol{h}}_v^{(l)}) = \frac{D[r_v^{(l)}]^2}{6B^2}$.

From the unbiased nature of quantization illustrated in Proposition 1, the calculated gradient is also unbiased. The approach imposes extra noise (i.e., the variance term in Proposition 1) to the gradient during the backward pass. From Proposition 1, the noise effect is inversely-correlated with the number of quantization bins $B$. To evaluate how the noise affects the model performance, we train three popular GNN models, namely, GCN, GraphSAGE, and GAT, on the *ogbn-arxiv* dataset (Hu et al., 2020) with different precisions (Table 1). The detailed experiment setting is elaborated in Appendix G.2. **Here we emphasize that all these three models are trained with full-batch data**.

Table 1: The test accuracy of GCN, GraphSAGE, and GAT trained on the *ogbn-arxiv* dataset with compressed activations storing in different precision. All results are averaged over ten random trials.

| Model | FP32 (Baseline) | INT8 | INT4 | INT2 | INT1 |
|---|---|---|---|---|---|
| GCN | 72.07±0.16 | 72.06±0.29 | 71.96±0.26 | 71.93±0.20 | 71.68±0.17 |
| GraphSAGE | 71.85±0.24 | 71.83±0.15 | 71.85±0.27 | 71.58±0.22 | 71.34±0.24 |
| GAT | 72.35±0.12 | 72.39±0.14 | 72.34±0.12 | 72.35±0.12 | 72.17±0.12 |

**Observations.** For all three models, the loss in accuracy is negligible ($\approx 0.2\%$) even using the vanilla 2-bit quantization. In contrast, for CNNs, adopting the vanilla INT1 or INT2 quantization will cause a significant accuracy drop (usually $> 5\%$) (Chen et al., 2021a; 2020a). This observation can be explained by the following mechanism. We show in Appendix E.4 that the approximation error will compound layer-by-layer during the backward pass. The layer-depth of GNNs is much smaller than CNNs due to the over-smoothing problem, and hence GNNs are much more noise-tolerant than CNNs. Our observation suggests that in practical scenarios, there is no need for an expensive sophisticated quantization approach (e.g., mixed precision quantization) as considered in previous works (Chen et al., 2021a; Fu et al., 2020).

## 3.2 STORING RANDOMLY PROJECTED ACTIVATIONS

Here we explore the direction of random projection. The key idea is to project the activation maps into a low-dimensional space that keeps the original information as much as possible. In this way, similar to the framework in Section 3.1, we only need to store the dimension reduced activation maps in memory. Figure 1b illustrates the workflow of GCN layers. During the forward pass, each layer's accurate activations (e.g., $\boldsymbol{h}_v^{(l)}$) are used to compute those for the subsequent layer. Then the accurate activations will be projected into a low-dimensional space, which can be expressed as:

$$\boldsymbol{h}_{v_{\text{proj}}}^{(l)} = \text{RP}(\boldsymbol{h}_v^{(l)}) = \boldsymbol{h}_v^{(l)}\boldsymbol{R}, \tag{5}$$

where $\boldsymbol{R} \in \mathbb{R}^{D \times R}$ is a random matrix ($R < D$) which satisfies $\mathbb{E}[\boldsymbol{R}\boldsymbol{R}^\top] = \boldsymbol{I}$. $\boldsymbol{H}_{\text{proj}}^{(l)} \in \mathbb{R}^{|\mathcal{V}| \times R}$ is the matrix containing all projected node embeddings $\boldsymbol{h}_{v_{\text{proj}}}^{(l)}$. In this paper, $\boldsymbol{R}$ is set as the **normalized Rademacher random matrix** (Achlioptas, 2001) due to its low sampling cost (detailed introduction in Appendix D). After projection, we store only $\boldsymbol{H}_{\text{proj}}^{(l)}$ in memory, and hence the compression ratio is $\frac{D}{R}$. During the backward pass, the projected node embeddings are inversely transformed by

$$\hat{\boldsymbol{h}}_v^{(l)} = \text{IRP}(\boldsymbol{h}_{v_{\text{proj}}}^{(l)}) = \boldsymbol{h}_{v_{\text{proj}}}^{(l)}\boldsymbol{R}^\top, \tag{6}$$

where $\text{IRP}(\cdot)$ is the inverse projection operation. $\hat{\boldsymbol{H}}^{(l)} = \boldsymbol{H}_{\text{proj}}^{(l)}\boldsymbol{R}\boldsymbol{R}^\top \in \mathbb{R}^{|\mathcal{V}| \times D}$ is the recovered activation map containing all $\hat{\boldsymbol{h}}_v^{(l)}$. The following proposition shows the effect of random projection.

**Proposition 2 (Proof in Appendix E)** *The above* RP *and* IRP *are unbiased operations, i.e.,* $\mathbb{E}[\hat{\boldsymbol{H}}^{(l)}] = \mathbb{E}[\text{IRP}(\text{RP}(\boldsymbol{H}^{(l)}))] = \boldsymbol{H}^{(l)}$. *For each* $\hat{\boldsymbol{h}}_v^{(l)}$ *in* $\hat{\boldsymbol{H}}^{(l)}$, *we have* $\text{Var}(\hat{\boldsymbol{h}}_v^{(l)}) = \frac{D-1}{R}||\boldsymbol{h}_v^{(l)}||_2^2$.

From the unbiased nature of random projection given in Proposition 2, the calculated gradient is also unbiased. The approach here also only imposes extra variance to the calculated gradient, where the variance linearly scales with the $\frac{D}{R}$ ratio. To quantitatively study the effect of the extra variance in scenarios of practical interest, we follow the same setting in Section 3.1 and show the performance

Table 2: The test accuracy of GCN, GraphSAGE, and GAT trained on the *ogbn-arxiv* dataset with randomly projected activations. All results are averaged over ten random trials.

| Model | Full-Dimension (Baseline) | $\frac{D}{R} = 2$ | $\frac{D}{R} = 4$ | $\frac{D}{R} = 8$ | $\frac{D}{R} = 16$ |
|---|---|---|---|---|---|
| GCN | 72.07±0.16 | 71.87±0.15 | 71.72±0.18 | 71.71±0.22 | 71.55±0.13 |
| GraphSAGE | 71.85±0.24 | 71.58±0.28 | 71.46±0.28 | 71.29±0.25 | 71.13±0.28 |
| GAT | 72.35±0.12 | 72.18±0.14 | 72.15±0.10 | 72.02±0.12 | 71.89±0.12 |

of GCN, GraphSAGE, and GAT trained on the *ogbn-arxiv* dataset with different $\frac{D}{R}$ ratios in Table 2.

**Observations.** We make two main observations: (1) For all three models, the loss in accuracy tends to range from negligible ($\approx 0.2\%$) to moderate ($\approx 0.5\%$) when $\frac{D}{R} \leq 8$. (2) Under the same compression ratio, the quantization is better than random projection in terms of the loss in accuracy (e.g., compare the $\frac{D}{R} = 16$ result in Table 2 with the INT2 result in Table 1).

### 3.3 EXACT: COMBINING RANDOM PROJECTION AND QUANTIZATION

We experimentally show that GNNs are noise-tolerant to inaccurate activations compressed by quantization and random projection. However, the highest compression ratio of these two methods is limited. The precision cannot be less than 1-bit for quantization and $\frac{D}{R}$ cannot surpass eight for random projection due to a large accuracy drop. However, GNNs' demand on memory is endless since real-world graphs can contain hundreds of millions of nodes and the graph size is ever growing. Motivated by this fact, we try to further push forward the memory saving as long as the loss of accuracy and the time overhead are both acceptable. We explore the direction of combining these two methods into one framework termed "EXACT" to maximize the memory saving. Specifically, we compress the activations and store only $\tilde{\boldsymbol{h}}_v^{(l)} = \mathrm{Quant}(\mathrm{RP}(\boldsymbol{h}_v^{(l)}))$ in memory during the forward pass. During the backward pass, the node embedding is recovered as $\hat{\boldsymbol{h}}_v^{(l)} = \mathrm{IRP}(\mathrm{Dequant}(\tilde{\boldsymbol{h}}_v^{(l)}))$. EXACT can achieve a superior compression ratio, e.g., the compression ratio of EXACT is roughly $128\times$ if $\frac{D}{R} = 8$ and the precision is INT2.

However, one practical question is *whether the variance will explode, leading to significantly worse performance when applying random projection and quantization sequentially.* Here we try to answer the above question both theoretically and experimentally. Below we first theoretically show that after applying random projection, the quantization range of projected node embeddings is bounded.

**Proposition 3 (Proof in Appendix E)** *For each projected node embedding* $\boldsymbol{h}_{v_{\mathrm{proj}}}^{(l)} = \mathrm{RP}(\boldsymbol{h}_v^{(l)}) = \boldsymbol{h}_v^{(l)}\boldsymbol{R}$, *for* $\forall \epsilon > 0$, *by choosing* $s = ||\boldsymbol{h}_v^{(l)}||_2\sqrt{\frac{2\ln(2R/\epsilon)}{R}}$, *we have* $P(||\boldsymbol{h}_{v_{proj}}^{(l)}||_\infty \leq s) \geq 1 - \epsilon$.

From Proposition 3, after projection, the maximal absolute value among weights in $\boldsymbol{h}_{v_{\mathrm{proj}}}^{(l)}$ is bounded. Following the fact that the quantization range $r_{v_{\mathrm{proj}}}^{(l)} = \max\{\boldsymbol{h}_{v_{\mathrm{proj}}}^{(l)}\} - \min\{\boldsymbol{h}_{v_{\mathrm{proj}}}^{(l)}\} \leq 2||\boldsymbol{h}_{v_{\mathrm{proj}}}^{(l)}||_\infty$, $r_{v_{\mathrm{proj}}}^{(l)}$ is also bounded. Recall that the variance of quantization scales with $\frac{[r_{v_{\mathrm{proj}}}^{(l)}]^2}{B^2}$ (Proposition 1). Applying random projection and quantization sequentially will not lead to the variance explosion.

We also experimentally visualize the infinity norm of projected embeddings in Appendix E Figure 4. The infinity norm of projected embeddings may be even less than those of original embeddings when $R$ is larger than a threshold ($R = 0.5D$ in Figure 4). Thus, the extra variance is expected to be dominated by random projection, and hence the loss in accuracy is largely determined by the $\frac{D}{R}$ ratio. We also experimentally investigate the effect of EXACT by plotting the test accuracy against each quantization precision and the $\frac{D}{R}$ ratio.

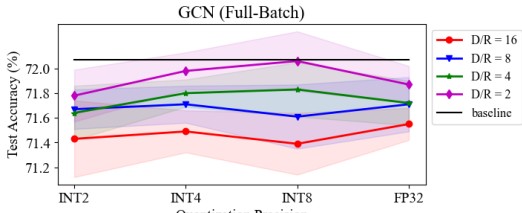

Figure 2: The performance of EXACT is mainly determined by the $\frac{D}{R}$ ratio of random projection.

Due to the page limit, we only present one representative result in Figure 2, namely, training GCNs on *ogbn-arxiv* with EXACT. More similar results can be found in Appendix I.1 Here we do not

consider INT1 precision since its model performance drop is already near 0.5%. We summarize two main observations. First, the performance of EXACT is largely determined by the $\frac{D}{R}$ ratio of random projection. Second, when $\frac{D}{R} \leq 8$, the loss in accuracy generally ranges from $\approx 0.2\%$ to $\approx 0.5\%$, regardless of the quantization precision.

**System Implementation.** In EXACT, each operation (e.g., SPMM, ReLU, and BatchNorm) is re-implemented with different configurations using CUDA kernels. For example, BatchNorm only supports quantization, while random projection is not applicable to it. Since Pytorch only supports precision down to INT8, we convert quantized tensors into bit-streams by CUDA kernels to maximize the memory saving. The configuration and implementation details are given in Appendix F.

## 4  RELATED WORK AND DISCUSSION

Due to the page limit, we briefly review and discuss the relationship between existing works and EXACT. A more comprehensive discussion can be found in Appendix B. Existing works can be roughly divided into *scalable/efficient GNN inference* and *scalable GNN training* according to the problem they try to solve. Almost all previous GNN quantization works try to solve the first problem, which is much simpler than the second problem that EXACT tries to address. Specifically, most of them try to enable the usage of low precision integer arithmetic during inference (Tailor et al., 2021; Zhao et al., 2020) by simulating the quantization effect, which may even increase the memory consumption during training. Feng et al. (2020) tries to address the second problem by proposing a heterogeneous quantization framework which assigns different bits to node embeddings in each layer. However, this framework is impractical on off-the-shell hardwares. For scalable GNN training methods, EXACT is orthogonal to most of them, including distributed training (Zheng et al., 2020b; Jia et al., 2020; Zhu et al., 2019; Wan et al., 2021), subgraph sampling (Hamilton et al., 2017; Zeng et al., 2020; Chiang et al., 2019), and historical embedding-based methods (Fey et al., 2021). We discuss the potential benefit of integrating EXACT with them in Appendix B.

## 5  EXPERIMENTS

The experiments are designed to answer the following research questions. **RQ1**: How effective is EXACT in terms of model performance at different compression rates (Section 5.1)? **RQ2**: Is the training process of deeper GNNs also robust to the noise introduced by EXACT (Section 5.1)? **RQ3**: How sensitive is EXACT to its key hyperparameters (Appendix I.4)? **RQ4**: What is the running time overhead of EXACT (Section 5.2)? **RQ5**: Is the convergence speed of GNNs impacted by EXACT (Appendix I.3)? **RQ6**: To what extent can EXACT reduce the hardware requirement for training GNNs on large graphs (Section 5.3)?

**Datasets and Models.** To evaluate the scalability of EXACT, we adopt five common large-scale graph benchmark datasets from different domains. Namely, Reddit, Flickr, Yelp, *ogbn-arxiv*, and *ogbn-products*. We evaluate EXACT under both the mini-batch training and full-batch training settings. In the mini-batch training setting, we integrate EXACT with two state-of-the-art subgraph sampling methods, namely Cluster-GCN (Chiang et al., 2019) and GraphSAINT (Zeng et al., 2020). In the full-batch training setting, we integrate EXACT with three popular models, including two commonly used shallow models, namely GCN (Kipf & Welling, 2017) and GraphSAGE (Hamilton et al., 2017), and one deep model GCNII (Chen et al., 2020b). To avoid confusions, **GCN, Graph-SAGE, and GCNII are both trained with the whole graph at each step**. For a fair comparison, we use the mean aggregator for GraphSAGE, Cluster-GCN, and GraphSAINT throughout the paper. Details about the hyperparameters of models and datasets can be found in Appendix G.

**Hyperparameter Settings.** From Section 3.3, we show that the performance of EXACT is largely determined by the $\frac{D}{R}$ ratio of random projection, and the model performance drop with INT2 quantization is negligible. Thus, to balance the accuracy drop and memory saving ratio, we adopt INT2 precision with different $\frac{D}{R}$ ratios for EXACT. Specifically, EXACT (INT2) indicates that it only applies 2-bit quantization to the activation maps of all applicable operations (see Table 8). EX-ACT(RP+INT2) indicates that it applies random projection followed by a 2-bit quantization to the activations of all applicable operations. We perform a grid search for $\frac{D}{R}$ ratio from {2, 4, 8}. Detailed $\frac{D}{R}$ configuration for EXACT(RP+INT2) can be found in Table 12.

Table 3: Comparison on the test accuracy/F1-micro and memory saving on five datasets. The hardware here is a single RTX 3090 (24GB). "Act Mem." is the memory (MB) occupied by activation maps. "OOM" indicates the out-of-memory error. **Bold faces** indicate that the loss in accuracy is negligible ($\approx 0.2\%$) or the result is better compared to the baseline. Underline numbers indicate that the loss in accuracy is moderate ($\approx 0.5\%$). All reported results are averaged over ten random trials.

| | # nodes | 230K | | 89K | | 717K | | 169K | | 2.4M | |
|---|---|---|---|---|---|---|---|---|---|---|---|
| | # edges | 11.6M | | 450K | | 7.9M | | 1.2M *ogbn-arxiv* | | 61.9M *ogbn-products* | |
| Model | Methods | Reddit | | Flickr | | Yelp | | | | | |
| | | Acc. | Act Mem. | Acc. | Act Mem. | F1-micro | Act Mem. | Acc. | Act Mem. | Acc. | Act Mem. |
| Cluster-GCN | Baseline | 95.62±0.10 | 14.5 (1×) | 49.61±0.47 | 16.5 (1×) | 63.98±0.14 | 29.3 (1×) | — | — | 78.62±0.26 | 35.2 (1×) |
| | EXACT(INT2) | 95.58±0.09 | 2 (7.3×) | 49.69±0.20 | 1.5 (11×) | 63.90±0.15 | 4 (7.3×) | — | — | 78.47±0.40 | 2.5 (14×) |
| | EXACT(RP+INT2) | 95.32±0.07 | 1.4 (10.4×) | 49.31±0.21 | 0.9 (18.3×) | 63.61±0.21 | 3 (9.8×) | — | — | 77.86±0.28 | 2.2 (16×) |
| Graph-SAINT | Baseline | 96.02±0.08 | 44.3 (1×) | 51.11±0.28 | 88.7 (1×) | 63.78±0.12 | 33.5 (1×) | 71.49±0.20 | 270 (1×) | 79.03±0.23 | 516 (1×) |
| | EXACT(INT2) | 95.96±0.05 | 6.6 (6.7×) | 50.86±0.32 | 7.8 (11.4×) | 63.77±0.14 | 4.3 (7.8×) | 71.76±0.15 | 20 (13.5×) | 78.94±0.28 | 40.5 (12.7×) |
| | EXACT(RP+INT2) | 95.69±0.06 | 3.6 (12.3×) | 50.65±0.17 | 3.4 (26×) | 63.41±0.19 | 3.3 (10.2×) | 71.44±0.16 | 10.8 (25×) | 78.50±0.41 | 29.5 (17.5×) |
| GCN | Baseline | 95.39±0.04 | 1029 (1×) | 53.08±0.14 | 378.8 (1×) | 40.22±0.47 | 6429 (1×) | 72.07±0.16 | 729.4 (1×) | — | — |
| | EXACT(INT2) | 95.36±0.03 | 122.8 (8.4×) | 52.92±0.20 | 37 (10.2×) | 40.20±0.38 | 640 (10×) | 72.04±0.21 | 54.5 (13.4×) | — | — |
| | EXACT(RP+INT2) | 95.30±0.03 | 67 (15.4×) | 52.90±0.22 | 17.8 (21.3×) | 39.89±0.56 | 427 (15×) | 71.67±0.16 | 30.2 (24.1×) | — | — |
| Graph-SAGE | Baseline | 96.44±0.04 | 1527.4 (1×) | 51.74±0.13 | 546.9 (1×) | 62.05±0.14 | 6976 (1×) | 71.85±0.24 | 786.2(1×) | 78.78[2]±0.19 | OOM |
| | EXACT(INT2) | 96.40±0.05 | 155.5 (9.8×) | 51.97±0.22 | 49.3 (11.1×) | 61.95±0.12 | 680 (10.3×) | 71.71±0.38 | 60.8 (12.9×) | 78.79±0.12 | 1144 |
| | EXACT(RP+INT2) | 96.34±0.03 | 71.7 (21.3×) | 51.83±0.21 | 20.4 (26.8×) | 61.59±0.12 | 466.5 (15×) | 71.32±0.26 | 30.5 (25.8×) | 78.76±0.13 | 572 |
| GCNII | Baseline | 96.71±0.07 | 5850 (1×) | 54.23±0.77 | 4067 (1×) | OOM | OOM | 72.85±0.27 | 14409 (1×) | — | — |
| | EXACT(INT2) | 96.65±0.06 | 388 (15×) | 54.55±0.47 | 256.4 (15.9×) | 64.79±0.09 | 2236 | 72.85±0.43 | 899 (16×) | — | — |
| | EXACT(RP+INT2) | 96.51±0.09 | 197.8 (29.6×) | 53.96±0.58 | 127.6 (31.9×) | 64.01±0.17 | 1649 | 72.67±0.45 | 451.2 (32×) | — | — |

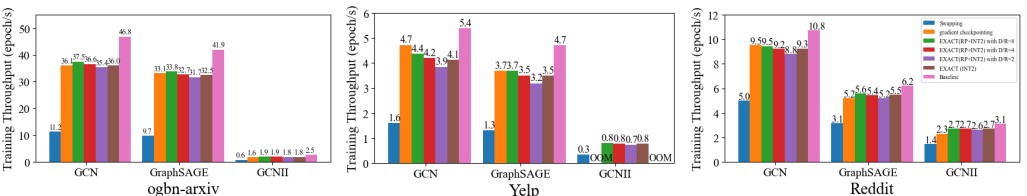

Figure 3: Training throughput comparison on a single RTX 3090 (24GB) GPU (higher is better). The time overhead of EXACT is roughly $12\%$-$25\%$. We discuss about swapping and gradient checkpointing in Appendix I.2.

## 5.1 TRADE-OFF BETWEEN SPACE AND MODEL PERFORMANCE

To answer **RQ1** and **RQ2**, we first analyze the trade-off between the model performance and memory saving. Table 3 summarizes the model performance and the memory footprint of the activation maps. Besides the memory usage of activation maps, a detailed analysis about the overall memory saving ratio is provided in Appendix H. We make two main observations.

**First, EXACT aggressively reduces the memory footprint with $\leq 0.5\%$ loss in accuracy.** From Table 3, EXACT (INT2) reduces the memory footprint of activation maps from $7\times$ to $16\times$, with negligible loss in accuracy ($\approx 0.2\%$) in almost all experiments. EXACT (RP+INT2) generally reduces the memory footprint of activation maps from $10\times$ to $32\times$, and the loss in accuracy tends to range from lossless ($0.0\%$) to moderate ($\approx 0.5\%$). Although EXACT can aggressively compress the memory usage of activations associated with MM and SPMM by up to $128\times$, the compression ratio for other operations usually cannot surpass $32\times$ (e.g., the activation maps of ReLU and Dropout take exact one bit to store, see Appendix F). As a result, the overall compress ratio is pulled down. We show in Appendix H that the measured compression ratio is consistent with the theoretical one.

**Second, the training process of deeper models is also robust to extremely compressed activations.** As we analyzed in Appendix E.4, the vanilla quantization and random projection obtain unreasonable good performance might because the commonly used GNNs are shallow. Thus, one open question is that, *to what extent can deeper GNNs robust to extremely compressed activation maps* (**RQ2**)? The ablation study of GCNII in Table 3 tries to experimentally answer this question. The training process of deeper GNNs (up to 16-layer, see Table 17) is also robust to the noise introduced by the extremely compressed activations, with up to $32\times$ less memory footprint of activation maps. We note that building deeper GNNs is still an open direction due to the over-smoothing and the extensive memory consumption problem (Oono & Suzuki, 2019; Li et al., 2021; Chen et al., 2020b). From the practical usage perspective, our GCNII ablation study indicates that EXACT may enable to train deeper GNNs on large graphs with minimal loss in performance.

---

[2]The regular training raises OOM error. The result is obtained using the automated mixed-precision training (torch.cuda.amp).

To answer **RQ3**, we present the sensitivity study of EXACT (RP+INT2) to $\frac{D}{R}$ ratio in Appendix I.4. In general, the model performance drop of EXACT (RP+INT2) increases with the $\frac{D}{R}$ ratio. In summary, when $\frac{D}{R} = 8$, the loss in accuracy of EXACT (RP+INT2) is $\leq 0.5\%$ in roughly two-third of experiments and $\leq 1\%$ in almost all experiments.

## 5.2 TRADE-OFFS BETWEEN SPACE AND SPEED

To answer **RQ4**, we compare the training throughput of EXACT with the baseline using a single RTX 3090 (24GB) GPU. Figure 3 shows the results among EXACT with different configurations and the baseline (see Table 9 for detailed package information). We make two main observations.

**First, the overhead of EXACT (INT2) is roughly** $12\% \sim 25\%$**.** The overhead of EXACT (INT2) comes entirely from quantization. We note that EXACT (INT2) adopts vanilla integer quantization with negligible loss in accuracy, which has the lowest overhead among all quantization methods. Sophisticated quantization strategies (e.g., mixed precision quantization and non-uniform quantization) further increase the time overhead, because they require an extra search process to find precision or quantization step for each layer/sample (Chakrabarti & Moseley, 2019; Fu et al., 2020; Chen et al., 2021a). From Figure 3, the overhead of EXACT is roughly $12\% \sim 25\%$.

**Second, the running speed of EXACT (RP+INT2) is comparable or even faster than quantization only.** The above counter-intuitive observation can be explained by the following mechanism. Although random projection introduces two extra matrix multiplication operations, the total number of elements to be quantized is $\frac{D}{R}$ times less than directly quantizing the activations. Considering that quantization is the main bottleneck, compared to EXACT (INT2), EXACT (RP+INT2) achieves roughly twice overall compression ratios with a comparable or even smaller time overhead.

**RQ4** focuses on the actual running speed measured by the hardware throughput. Here we investigate the convergence speed from the optimization perspective (**RQ5**). To answer **RQ5**, due to the page limit, we present the training curve of models trained on Reddit dataset with EXACT in Appendix I.3. The converge speed here is measured by the number of epochs it takes to reach a certain validation accuracy. In summary, the convergence speed of EXACT is slightly slower than the baseline, with limited impact on final accuracy.

## 5.3 ABLATION STUDIES

**Train full-batch GraphSAGE on** *ogbn-products* **using a GPU with 11GB memory.**

Table 3 shows that EXACT reduces the memory footprint of activation maps by up to $32\times$. However, as analyzed in Appendix H, besides the activation maps, input data (including feature matrices $X$, adjacency matrix $A$, and labels), activation gradients also occupy memory in the full-batch training. For *ogbn-products*, the input data occupies 4.7GB (see Appendix H). From the OGB leader-board, training a full-batch GraphSAGE re-

Table 4: The test accuracy of full-batch GraphSAGE trained on *ogbn-products* using a single GTX 1080 Ti (11GB).

| Methods | Accuracy |
|---|---|
| AMP | OOM |
| EXACT(RP+INT2) + AMP | 78.28±0.18 |

quires GPUs with at least 48GB memory. We note that EXACT can be integrated with AMP (see Appendix F) to further squeeze out the memory. To answer **RQ6**, by integrating AMP and EXACT (RP+INT2) with $\frac{D}{R} = 4$, we successfully train a full-batch GraphSAGE on *ogbn-products* on a single GTX 1080 Ti with 11GB memory, with moderate loss ($\approx 0.5\%$) in accuracy (Table 4).

## 6 CONCLUSION AND FUTURE WORK

In this paper, we propose EXACT, a simple-yet-effective framework for training GNNs with compressed activations. We demonstrate the potential of EXACT for the real-world usage by systematically evaluating the trade-off among the memory-saving, time overhead, and accuracy drop. We show that EXACT is orthogonal to most of the existing solutions and discuss how EXACT can be integrated with them in Appendix B. Future work includes (1) evaluating EXACT under multi-GPU settings for distributed training; (2) combining EXACT with historical embedding-based methods; (3) combining EXACT with memory-efficient training systems, such as swapping.

## 7 Acknowledgement

We would like to thank all the anonymous reviewers for their valuable suggestions. Thank all the members of Samsung Research America advertisement intelligence team for your feedback, and everyone who has provided their generous feedback on this work. This work is, in part, supported by NSF IIS-1750074. The views and conclusions contained in this paper are those of the authors and should not be interpreted as representing any funding agencies.

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

# A  NOTATIONS

Table 5: Table of Notations.

| Notations | Description |
|---|---|
| $|\mathcal{V}|$ | The number of nodes |
| $|\mathcal{E}|$ | The number of edges |
| $\boldsymbol{A} \in \mathbb{R}^{|\mathcal{V}| \times |\mathcal{V}|}$ | The adjacency matrix, which is stored in sparse matrix format |
| $\hat{\boldsymbol{A}} \in \mathbb{R}^{|\mathcal{V}| \times |\mathcal{V}|}$ | The normalized adjacency matrix, which is stored in sparse matrix format |
| $\boldsymbol{X}$ | The input feature matrix |
| $\boldsymbol{h}_v^{(l)}$ | The uncompressed node embedding at the $l^{\text{th}}$ layer corresponding to node $v$ |
| $\boldsymbol{H}^{(l)}$ | The uncompressed node embedding matrix at the $l^{\text{th}}$ layer (each row is a node embedding $\boldsymbol{h}_v^{(l)}$ ) |
| $D$ | The node embedding dimension |
| $\boldsymbol{h}_{v_{\text{INT}}}^{(l)}$ | The quantized node embedding at the $l^{\text{th}}$ layer corresponding to node $v$ |
| $\boldsymbol{H}_{\text{INT}}^{(l)}$ | The quantized node embedding matrix at the $l^{\text{th}}$ layer (each row is a $\boldsymbol{h}_{v_{\text{INT}}}^{(l)}$) |
| $r_v^{(l)}$ | The quantization range, where $r_v^{(l)} = \max\{\boldsymbol{h}_v^{(l)}\} - \min\{\boldsymbol{h}_v^{(l)}\}$ |
| $b$ | The bit-width. In this paper, $b$ must be chosen from $\{1, 2, 4, 8, 32\}$ |
| $\boldsymbol{h}_{v_{\text{proj}}}^{(l)}$ | The randomly projected node embedding at the $l^{\text{th}}$ layer corresponding to node $v$ |
| $\boldsymbol{H}_{\text{proj}}^{(l)}$ | The randomly projected node embedding matrix at the $l^{\text{th}}$ layer (each row is a $\boldsymbol{h}_{v_{\text{proj}}}^{(l)}$) |
| $R$ | The dimension of projected node embeddings |
| $\hat{\boldsymbol{h}}_v^{(l)}$ | The recovered node embedding at the $l^{\text{th}}$ layer corresponding to node $v$ (either compressed by quantization or random projection) |
| $\hat{\boldsymbol{H}}^{(l)}$ | The recovered node embedding matrix at the $l^{\text{th}}$ layer (each row is a $\hat{\boldsymbol{h}}_v^{(l)}$ ) |
| $\text{Var}(\boldsymbol{h})$ | The variance of the vector $\boldsymbol{h}$, where $\text{Var}(\boldsymbol{h}) = \mathbb{E}[\boldsymbol{h}^\top \boldsymbol{h}] - (\mathbb{E}[\boldsymbol{h}])^\top (\mathbb{E}[\boldsymbol{h}])$ |
| $||\boldsymbol{h}||_2$ | The 2-norm of vector $\boldsymbol{h}$, where $||\boldsymbol{h}||_2 = \sqrt{\boldsymbol{h}^\top \boldsymbol{h}}$ |
| $||\boldsymbol{h}||_\infty$ | The infinity norm of vector $\boldsymbol{h}$, where $||\boldsymbol{h}||_\infty = \max |\boldsymbol{h}|$ |

# B  RELATED WORK AND DISCUSSION

Due to the page limit, we discuss the relationship between EXACT and existing works in detail here. GNN is notoriously inefficient and non-scalable during both the training and inference phase. Previous works can be divided into two categories according to the problem they try to solve. Namely, scalable/efficient GNN inference and scalable GNN training. Below we introduce and discuss the relationship between them and EXACT.

## B.1  SCALABLE/EFFICIENT GNN INFERENCE

Scalable GNN inference is an orthogonal direction to the problem that EXACT tries to address. Below we introduces two popular directions for scalable GNN inference. Namely, GNN quantization and graph sparsification.

**GNN Quantization.** Most of the previous GNN quantization works focus on enabling the usage of low precision integer arithmetic during inference. Specifically, Tailor et al. (2021) proposes a Quantization Aware Training method tailored for GNNs, which emulates inference-time quantization during the training phase. Zhao et al. (2020) proposes to jointly search the quantization bit-width and GNN architectures. Note that both of them do not actually convert the node embeddings into lower numerical precision during training. Instead, they use full precision data type to simulate the effect of real quantization. Feng et al. (2020) proposes a heterogeneous quantization framework which assigns different bits to node embeddings in each layer while maintaining the weights at full precision. However, due to the mismatch in operands' bit-width, it is impractical to use in general purpose hardwares, such as CPUs and GPUs.

**Graph Sparsification.** The adjacency matrices are usually stored as sparse matrices. When hardwares perform SPMM , the sparse matrix format leads to random memory accesses and limited data reuse due to its irregular structure. Thus, GNNs have higher inference latency than other neural networks. Graph sparsification can alleviate the issue by removing redundant edges from original graphs. Zheng et al. (2020a) proposes a learning-based graph sparsification method which removes

potentially task-irrelevant edges from input graphs Li et al. (2020) formulates the graph sparsification problem as an optimization objective which can be solved by alternating direction method of multipliers (ADMM). Chen et al. (2021b) proposes to co-simplify the input graph and GNN model by extending the iterative magnitude pruning to graph areas.

### B.2 SCALABLE GNN TRAINING

In this subsection, we introduce previous works that try to train GNNs on large graphs. **We conclude that EXACT are parallel to most of the previous works.**

**Subgraph Sampling/Mini-Batch Training.** As we introduced in the main body, most of the previous scalable GNN training methods fall into this category. The key idea is to train GNNs with sampled subgraphs instead of the whole graph at each step. In this way, only node embeddings that are present in the current subgraph will be retained in memory. Based on this idea, various sampling techniques have been proposed, including the node-wise sampling (Hamilton et al., 2017; Chen et al., 2017; Markowitz et al., 2021), layer-wise sampling (Zou et al., 2019; Chen et al., 2018; Huang et al., 2018), and subgraph sampling (Chiang et al., 2019; Zeng et al., 2020). Generally, methods in this category are orthogonal to EXACT, and they can certainly be combined. Similar to EXACT, subgraph sampling also introduce the gradient noise during the training process. However, the gradient noise does not necessarily result in the model performance drop. In contrast, many previous works experimentally show that subgraph sampling based methods may even yield better performance than the whole graph training (Zeng et al., 2020; Hu et al., 2020). Our experiment results also exhibit a similar phenomenon that EXACT may achieve better performance than the original one. This observation can be explained by the regularization effect introduced by the gradient noise (Hu et al., 2020).

In this paper, we show in experiments that EXACT can be combined with subgraph-sampling based methods with only limited accuracy drop. From the practical usage perspective, EXACT can enlarge the maximal subgraph size for subgraph training-based method. As a result, the error introduced by EXACT can be compensated by using larger subgraphs, which previously cannot fit into GPUs. We leave it as one future direction.

**GNNAutoScale (Fey et al., 2021).** GNNAutoScale follows the idea of utilizing historical embeddings from prior training iterations (Fey et al., 2021). Specifically, it stores a historical embedding for each node as an offline storage in CPU memory. At each training step, it first samples a mini-batch of nodes. Then for out-of-mini-batch nodes, GNNAutoScale swaps their historical embeddings from CPU memory to GPU memory and utilizes their non-trainable historical embeddings for propagation. In this way, only the node embeddings inside the current mini-batch and those of their direct 1-hop neighbors are retained in memory. The time overhead of GNNAutoScale is mainly from the swapping step. EXACT is orthogonal to GNNAutoScale. It would be interesting to store node embeddings compressed by EXACT as historical embeddings for GNNAutoScale. Considering that quantization is more time-efficient compared to swapping, the time overhead of GNNAutoScale can be greatly reduced since the time cost of swapping scales with the total number of bits to be swapped.

**Distributed Training.** Unlike the data in other domains, the graph data cannot be trivially divided into mini-batches due to its connectivity between samples. The graph distributed training methods require to split the graph into mini-batches that minimizes the communication overhead. Specifically, AliGraph (Zhu et al., 2019) is a distributed GNN framework on CPU platforms, which does not support GPUs acceleration. ROC (Jia et al., 2020) learns to optimize the graph partitioning by predicting the execution time of performing a GNN operation on an input subgraph. PipeGCN (Wan et al., 2022) proposes to train GNNs with stale features and stale feature gradients under the distributed training setting. Both BNS-GCN (Wan et al., 2021) and DistDGL (Zheng et al., 2020b) leverage METIS (Karypis & Kumar, 1998) to performance the graph partitioning to minimize the communication cost. However, one main drawback of applying METIS is that the ahead-of-training overhead of METIS is relatively large on huge graphs. EXACT is also orthogonal to the distributed training methods and can be used to decrease the hardware requirements. It would be interesting to investigate the trade-off among the accuracy drop and the memory saving of EXACT under multi-GPU settings.

**Memory efficient training system.** Gradient checkpointing (Chen et al., 2016; Jain et al., 2019; Kirisame et al., 2020; Shah et al., 2020) trades computation for memory by dropping some of the

---

**Algorithm 1:** Forward Pass of the $l^{\text{th}}$ GCN layer

---

**Input:** $\boldsymbol{H}^{(l)}, \boldsymbol{\Theta}^{(l)}$, the total number of layers $L$.
**Output:** $\boldsymbol{H}^{(l+1)}$

```
1  ctx^(l) ← {} /* the context which saves tensors for backward              */
```
2  $\boldsymbol{J}^{(l)} \leftarrow \texttt{MM}(\boldsymbol{H}^{(l)}, \boldsymbol{\Theta}^{(l)})$
```
3  Add H^(l) and Θ^(l) to ctx^(l)  /* used for MM.backward                   */
```
4  $\boldsymbol{H}^{(l+1)} \leftarrow \texttt{SPMM}(\hat{\boldsymbol{A}}, \boldsymbol{J}^{(l)})$
```
5  Add Â (in CSR format) and J^(l) to ctx^(l)   /* used for SPMM.backward    */
```
6  **if** $l \neq L - 1$ **then**
```
7  │   Add 1_{H^(l+1)>0} to ctx^(l) /* used for ReLU.backward               */
```
8  │   $\boldsymbol{H}^{(l+1)} = \text{ReLU}(\boldsymbol{H}^{(l+1)})$
9  **end**
10 **return** $\boldsymbol{H}^{(l+1)}$

---

activations in the forward pass and recomputing them in the backward pass. Swapping (Meng et al., 2017) utilizes the huge amount of available CPU memory by swapping tensors between CPU and GPU. EXACT is also parallel to both the gradient checkpointing and swapping. As a result, EXACT can be combined with them to further squeeze out the memory, at the cost of larger time overhead.

**Other Activation Compression Techniques.** JPEG-ACT (Evans et al., 2020) extends the widely-used JPEG method to compress the activations. Cusz is a prediction-based lossy compression method with GPU support (Tian et al., 2020; Tao et al., 2017). Both of them can be applied to compress the activations of GNNs. However, one practical problem is that they have hyperparameters which are hard to tune. For example, in JPEG-ACT, there is a hyperparameter $\alpha$ controls the space/accuracy trade-off. This problem can be alleviated by incorporating one recent proposed method AC-GC (Evans & Aamodt, 2021). Given a preset allowable increase in loss, AC-GC can automatically find the hyperparameter such that the compression rate can adapt to the preset training conditions. We note that EXACT and AC-GC approaches the problem from different perspective. For EXACT, we value the compression rate more than the accuracy drop. Namely, given a preset memory budget, we try to fit the model with acceptable accuracy drop. For AC-GC, they value the accuracy change more than the compression rate in the sense that the compression rate is adapted to the given allowable accuracy drop.

## C    ANALYZING THE MEMORY CONSUMPTION OF GCNS

Generally, each training step contains a forward pass phase and a backward pass phase. During the forward pass, activation maps of each operation are kept in memory for computing the gradients. During the backward pass, the weight gradients and activation gradients are calculated and stored in memory. For weight gradients, their memory usage is negligible since they share the same shape with model weights. For activation gradients, their memory usage is dynamic and will eventually be cleared to zero. In standard deep learning libraries, activation gradients will be deleted as soon as their reference counting becomes zero (Paszke et al., 2019; Abadi et al., 2016). Thus, storing activation maps take up most of the memory. The memory usage of activation maps in GCN layers are given in Algorithm 1.

## D    RANDOM PROJECTION

In this paper, we use the normalized Rademacher random matrix (Achlioptas, 2001) for projecting the activation maps. Specifically, for a $D \times R$ normalized Rademacher random matrix $\boldsymbol{R}$, each element in $\boldsymbol{R}$ is i.i.d. sampled from the following distribution:

$$\boldsymbol{R}_{i,j} = \sqrt{\frac{1}{R}} \times \begin{cases} +1, & \text{with probability } \frac{1}{2}, \\ -1, & \text{with probability } \frac{1}{2}. \end{cases} \tag{7}$$

where $\sqrt{\frac{1}{R}}$ is the normalize factor such that $\mathbb{E}[\boldsymbol{R}\boldsymbol{R}^\top] = \boldsymbol{I}$. From above, we can see that the sampling cost of the normalized Rademacher random matrix is relatively low since sampling from Bernoulli distribution is much cheaper than sampling from Gaussian distribution.

# E  THEORY

## E.1  PROOF OF PROPOSITION 1

**Proposition 1 (Details in Appendix E)** *The above quantization and dequantization are unbiased operations, i.e.,* $\mathbb{E}[\hat{\boldsymbol{h}}_v^{(l)}] = \mathbb{E}[\text{Dequant}(\text{Quant}(\boldsymbol{h}_v^{(l)}))] = \boldsymbol{h}_v^{(l)}$ *and* $\text{Var}(\hat{\boldsymbol{h}}_v^{(l)}) = \frac{D[r_v^{(l)}]^2}{6B^2}$.

Proposition 1 is adopted from the theoretical analysis in ActNN (Chen et al., 2021a). For completeness, we prove it here with more details. The conclusion of $\mathbb{E}[\hat{\boldsymbol{h}}_v^{(l)}] = \boldsymbol{h}_v^{(l)}$ follows from the fact that the stochastic rounding is an unbiased operation (Courbariaux et al., 2015). Specifically, $\forall$ scalar $h$, the stochastic rounding can be expressed as

$$\lfloor h \rceil = \begin{cases} \lceil h \rceil, & \text{with probability} \quad h - \lfloor h \rfloor, \\ \lfloor h \rfloor, & \text{with probability} \quad 1 - (h - \lfloor h \rfloor), \end{cases} \tag{8}$$

where $\lceil \cdot \rceil$ is the ceil operation and $\lfloor \cdot \rfloor$ is the floor operation. First, we have $\mathbb{E}[\lfloor h \rceil] = h$ because

$$\mathbb{E}[\lfloor h \rceil] = \lceil h \rceil(h - \lfloor h \rfloor) + \lfloor h \rfloor(1 - h + \lfloor h \rfloor)$$
$$= h \quad \text{(following the fact that } \lceil h \rceil - \lfloor h \rfloor = 1)$$

Hence,

$$\mathbb{E}[\hat{\boldsymbol{h}}_v^{(l)}] = \frac{r_v^{(l)}}{B}\mathbb{E}[\lfloor \frac{\boldsymbol{h}_v^{(l)} - Z_v^{(l)}}{r_v^{(l)}}B \rceil] + Z_v^{(l)} = \boldsymbol{h}_v^{(l)}.$$

Regarding the variance, let $\bar{\boldsymbol{h}} = \frac{\boldsymbol{h}_v^{(l)} - Z_v^{(l)}}{r_v^{(l)}}B = (h_1, \cdots, h_D)$. Suppose $\forall i, h_i - \lfloor h_i \rfloor = \sigma \sim$ Uniform$(0, 1)$, we have.

$$\text{Var}(\hat{\boldsymbol{h}}_v^{(l)}) = \frac{[r_v^{(l)}]^2}{B^2}\text{Var}(\lfloor \bar{\boldsymbol{h}} \rceil) = \mathbb{E}[\bar{\boldsymbol{h}}^\top \bar{\boldsymbol{h}}] - (\mathbb{E}[\bar{\boldsymbol{h}}])^\top(\mathbb{E}[\bar{\boldsymbol{h}}])$$

$$= \frac{[r_v^{(l)}]^2}{B^2}\sum_{i=1}^{D}\lceil h_i \rceil^2(h_i - \lfloor h_i \rfloor) + \lfloor h_i \rfloor^2(1 - h_i + \lfloor h_i \rfloor) - h_i^2$$

$$= \frac{[r_v^{(l)}]^2 D}{B^2}\left(2\lfloor h_1 \rfloor h_1 + h_1 - \lfloor h_1 \rfloor^2 - \lfloor h_1 \rfloor - h_1^2\right) \quad \text{(substitute } \lceil h_1 \rceil \text{ with } \lfloor h_1 \rfloor + 1)$$

$$= \frac{[r_v^{(l)}]^2 D}{B^2}\left(\sigma - \sigma^2\right) \quad \text{(substitute } \lfloor h_1 \rfloor \text{ with } h_1 - \sigma)$$

By taking expectation w.r.t. $\sigma$ on the both side, we have $\text{Var}(\hat{\boldsymbol{h}}_v^{(l)}) = \frac{[r_v^{(l)}]^2 D}{6B^2}$

$\square$

## E.2  PROOF OF PROPOSITION 2

**Proposition 2 (Proof in Appendix E)** *The above* RP *and* IRP *are unbiased operations, i.e.,* $\mathbb{E}[\hat{\boldsymbol{H}}^{(l)}] = \mathbb{E}[\text{IRP}(\text{RP}(\boldsymbol{H}^{(l)}))] = \boldsymbol{H}^{(l)}$. *For each* $\hat{\boldsymbol{h}}_v^{(l)}$ *in* $\hat{\boldsymbol{H}}^{(l)}$, *we have* $\text{Var}(\hat{\boldsymbol{h}}_v^{(l)}) = \frac{D-1}{R}||\boldsymbol{h}_v^{(l)}||_2^2$.

We can get $\mathbb{E}[\hat{\boldsymbol{H}}^{(l)}] = \mathbb{E}[\text{IRP}(\text{RP}(\boldsymbol{H}^{(l)}))] = \boldsymbol{H}^{(l)}$ directly following the fact that $\mathbb{E}[\text{IRP}(\text{RP}(\boldsymbol{H}^{(l)}))] = \mathbb{E}[\boldsymbol{H}^{(l)}\boldsymbol{R}\boldsymbol{R}^\top] = \boldsymbol{H}^{(l)}\mathbb{E}[\boldsymbol{R}\boldsymbol{R}^\top] = \boldsymbol{H}^{(l)}$.

Regarding the variance, let $\boldsymbol{P} = \boldsymbol{R}\boldsymbol{R}^\top \in \mathbb{R}^{D \times D}$. For the sake of notation convenience, we ignore the subscript and superscript in this subsection. Namely, we use the notation $\hat{\boldsymbol{h}}$ to represent $\hat{\boldsymbol{h}}_v^{(l)}$, and

use the notation $\boldsymbol{h}$ to represent $\boldsymbol{h}_v^{(l)}$. Also, we assume the shapes of $\hat{\boldsymbol{h}}$ and $\boldsymbol{h}$ are both $1 \times D$ in this subsection. We have

$$\begin{aligned}
\mathrm{Var}(\hat{\boldsymbol{h}}) &= \mathrm{Var}(\boldsymbol{h}\boldsymbol{P}) \\
&= \boldsymbol{h}\mathbb{E}[\boldsymbol{P}\boldsymbol{P}^\top]\boldsymbol{h}^\top - \boldsymbol{h}\boldsymbol{h}^\top
\end{aligned}$$

Next, we will characterize the distribution of $\boldsymbol{P}\boldsymbol{P}^\top$. Recall that $\boldsymbol{P} = \boldsymbol{R}\boldsymbol{R}^\top$, thus,

$$\boldsymbol{P}_{i,j} = \sum_{k=1}^{R} a_{i,k} a_{j,k},$$

where each $a_{i,j} = \pm\sqrt{\frac{1}{R}}$ with equal probability. Thus,

$$\begin{aligned}
[\boldsymbol{P}\boldsymbol{P}^\top]_{i,j} &= \sum_{m=1}^{D} \boldsymbol{P}_{i,m}\boldsymbol{P}_{j,m} \\
&= \sum_{m=1}^{D} \left( (\sum_{k=1}^{R} a_{i,k}a_{m,k})(\sum_{k=1}^{R} a_{j,k}a_{m,k}) \right).
\end{aligned} \tag{9}$$

From Equation 9, it is easy to show that $\mathbb{E}[\boldsymbol{P}\boldsymbol{P}^\top]$ is a diagonal matrix. Namely, $\forall i \neq j$, we have $\mathbb{E}[\boldsymbol{P}\boldsymbol{P}^\top]_{i,j} = 0$ following the fact that each $a_{i,j}$ are independent and $\mathbb{E}[a_{i,j}] = 0$. For elements on the diagonal, we have

$$\begin{aligned}
\mathbb{E}[\boldsymbol{P}\boldsymbol{P}^\top]_{i,i} &= \mathbb{E}[\sum_{m=1}^{D} \boldsymbol{P}_{i,m}\boldsymbol{P}_{i,m}] \\
&= \mathbb{E}[\sum_{m=1}^{D} (\sum_{k=1}^{R} a_{i,k}a_{m,k})^2] \\
&= \mathbb{E}[(\sum_{k=1}^{R} a_{i,k}a_{i,k})^2] + \mathbb{E}[\sum_{m\neq i} (\sum_{k=1}^{R} a_{i,k}a_{m,k})^2] \\
&= 1 + \sum_{m\neq i} \left( \sum_{k=1}^{R} \mathbb{E}[(a_{i,k}a_{m,k})^2] + 2 \sum_{p,q:p<q} \mathbb{E}[a_{i,p}a_{m,p}a_{i,q}a_{m,q}] \right) \\
&= 1 + \sum_{m\neq i} \left( \sum_{k=1}^{R} \mathbb{E}[(a_{i,k}a_{m,k})^2] \right) \\
&= 1 + \frac{D-1}{R}.
\end{aligned} \tag{10}$$

Let $\boldsymbol{h} = (h_1, \cdots h_D)$, we have

$$\begin{aligned}
\mathrm{Var}(\hat{\boldsymbol{h}}) &= \boldsymbol{h}\mathbb{E}[\boldsymbol{P}\boldsymbol{P}^\top]\boldsymbol{h}^\top - \boldsymbol{h}\boldsymbol{h}^\top \\
&= \boldsymbol{h} \begin{bmatrix} 1+\frac{D-1}{R} & & \\ & \ddots & \\ & & 1+\frac{D-1}{R} \end{bmatrix} \boldsymbol{h}^\top - \boldsymbol{h}\boldsymbol{h}^\top \\
&= \sum_{i=1}^{D} (1 + \frac{D-1}{R})h_i^2 - \sum_{i=1}^{D} h_i^2 \\
&= \frac{D-1}{R}||\boldsymbol{h}||_2^2.
\end{aligned} \tag{11}$$

$\square$

### E.3 PROOF OF PROPOSITION 3

**Proposition 3 (Proof in Appendix E)** *For each projected node embedding $\boldsymbol{h}_{v_{\text{proj}}}^{(l)} = \text{RP}(\boldsymbol{h}_v^{(l)}) = \boldsymbol{h}_v^{(l)}\boldsymbol{R}$, for $\forall \epsilon > 0$, by choosing $s = ||\boldsymbol{h}_v^{(l)}||_2 \sqrt{\frac{2\ln(2R/\epsilon)}{R}}$, we have $P(||\boldsymbol{h}_{v_{proj}}^{(l)}||_\infty \le s) \ge 1 - \epsilon$.*

For the normalized Rademacher random matrix $\boldsymbol{R} \in \mathbb{R}^{D \times R}$, we have $\boldsymbol{R}_{i,j} = \pm\frac{1}{\sqrt{R}}$ with equal probability. Let $\boldsymbol{h}_v^{(l)} = (h_1, \cdots, h_D)$ and $\boldsymbol{h}_{v_{\text{proj}}}^{(l)} = (u_1, \cdots, u_R)$. Since $\boldsymbol{h}_{v_{\text{proj}}}^{(l)} = \boldsymbol{h}_v^{(l)}\boldsymbol{R}$, we have $u_1 = \sum_{i=1}^D a_i h_i$, where $a_i = \pm\frac{1}{\sqrt{R}}$ with equal probability.

First, we have the following inequality:

$$
\begin{aligned}
\mathbb{E}[e^{tRu_1}] &= \prod_{i=1}^D \mathbb{E}[e^{tRa_i h_i}] \\
&= \prod_{i=1}^D \frac{e^{t\sqrt{R}h_i} + e^{-t\sqrt{R}h_i}}{2} = \prod_{i=1}^D \cosh t\sqrt{R}h_i \\
&\le \prod_{i=1}^D e^{\frac{t^2 R h_i^2}{2}} \quad \text{(following the fact that } \cosh x \le e^{\frac{x^2}{2}}\text{)} \\
&= e^{\frac{t^2 R ||\boldsymbol{h}_v^{(l)}||_2^2}{2}}.
\end{aligned}
\tag{12}
$$

For notation convenience, we use the notation "$C$" to represent the node embedding norm $||\boldsymbol{h}_v^{(l)}||_2$. For $\forall s, q > 0$, according to Chernoff bound, we have:

$$
\begin{aligned}
P(|u_i| \ge s) &= 2P(u_i \ge s) \\
&= 2P(e^{qu_i} \ge e^{qs}) \\
&\le 2\frac{\mathbb{E}[e^{qu_i}]}{e^{qs}} \quad \text{(by Chernoff bound)} \\
&\le 2\frac{\mathbb{E}[e^{\frac{sRu_i}{C^2}}]}{e^{\frac{s^2 R}{C^2}}} \quad \text{(by setting } q = \frac{sR}{C^2}\text{)} \\
&\le 2e^{-\frac{s^2 R}{2C^2}} \quad \text{(by Equation 12)}
\end{aligned}
\tag{13}
$$

Note that Equation 13 holds for $\forall s > 0$. Given a specific $\epsilon$, by setting $s = C\sqrt{\frac{2\ln(2R/\epsilon)}{R}}$, we have $P(|u_i| \ge s) \le \frac{\epsilon}{R}$. Also,

$$
\begin{aligned}
P(||\boldsymbol{h}_{v_{\text{proj}}}^{(l)}||_\infty \le s) &= \prod_{i=1}^R P(|u_i| \le s) \\
&= (1 - P(|u_i| \ge s))^R \\
&\ge (1 - \frac{\epsilon}{R})^R \\
&\ge 1 - \epsilon
\end{aligned}
$$

$\square$

Here we also experimentally verify Proposition 3 by visualizing the infinity norm of projected node embeddings in Figure 4. To avoid creating confusions, $\frac{D}{R} = 1$ in Figure 4 means we apply the random projection with $R = D$ on the activation maps. We can observe that in general, the infinity norm increases with the compression ratio $\frac{D}{R}$. However, we note that the infinity norm of projected embeddings may less than those of original embeddings when R is larger than a threshold. This suggests that when $R$ is below a certain threshold, quantizing projected embeddings only have limited influence on model performance. This claim is also verified in the main body of this paper.

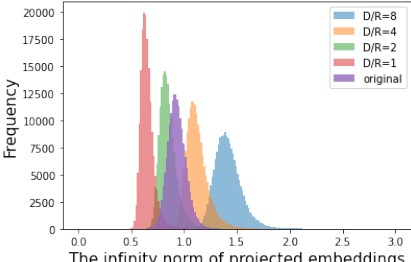 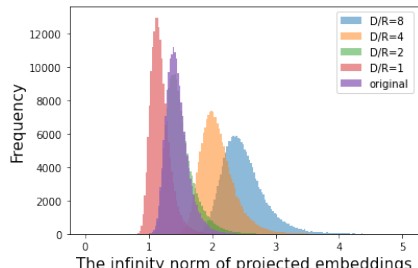

Figure 4: The histogram of the projected node embeddings' infinity norm at `MM` (the left figure) and `SPMM` (the right figure) operation of the first GCN layer.

### E.4 THE COMPOUND EFFECT OF APPROXIMATION ERRORS

In this subsection, we analyze why GNNs can tolerate the extremely compressed activations. From the numerical analysis (Bauer, 1974), the jacobian matrix (gradient) between two tensors can be viewed as a sum over all "paths" connecting these two tensors. Moreover, the architecture of GNNs is considerably shallower than CNNs and transformers. Informally, for GNNs, the key intuition is that these paths are significantly "shorter" compared to those of CNNs. As a result, the approximation error (i.e., the variance from quantization) along the path is hard to get accumulated.

We adopt Theorem 3 in ActNN (Chen et al., 2021a) to make the above statement more rigorous. Let $\hat{C}^{(m)}$ be the compressed context (either by quantization, random projection, or quantized random projection). Let $\nabla\Theta^{(l)}$ and $\nabla H^{(l)}$ be the gradient of $\Theta^{(l)}$ and $H^{(l)}$, respectively. $\hat{\nabla}\Theta^{(l)}$ and $\hat{\nabla}H^{(l)}$ are the calculated gradient using the compressed context, respectively. Further, we use the notation $G_\Theta^{(l\sim m)}(\hat{\nabla}H^{(m)}, \hat{C}^{(m)})$ to represent the variance introduced by utilizing the compressed context $\hat{C}^{(m)}$. Specifically, for a $L$ layer GNN, we have

$$\mathrm{Var}(\hat{\nabla}\Theta^{(l)}) = \mathrm{Var}(\nabla\Theta^{(l)}) + \sum_{m=l}^{L} \mathbb{E}\left[\mathrm{Var}\left(G_\Theta^{(l\sim m)}(\hat{\nabla}H^{(m)}, \hat{C}^{(m)})|\hat{\nabla}H^{(m)}\right)\right], \qquad (14)$$

where $\mathrm{Var}(\cdot|\hat{\nabla}H^{(m)})$ is the conditional variance, and $\mathrm{Var}(\nabla\Theta^{(l)})$ is the variance is from the mini-batch sampling variance. The key insights from Equation 14 are two folds. First, since the variances introduced by compressed contexts at different layers will accumulate, it suggests that the noise introduced by the compressed context is relatively small for shallow models. Considering that most of GNNs are usually less than four layers, this may explain why the loss in accuracy is negligible when using the vanilla INT2 quantization and a relatively large $\frac{D}{R}$ ratio. Second, under the subgraph training setting, the extra variance introduced by quantization can be compensated when using larger batch size (i.e., the size of subgraph in the graph learning).

In Table 1, we evaluate GNNs under the full-batch setting. Therefore, in addition to the reason of mentioned shallower architecture, it is also possible that the activation compressed training benefits from the large batch size, which leads to a much smoother gradient. To quantitatively study the effect of the batch size, below we present the ablation studies of two representative sampling based methods with much smaller batch size. Namely, GraphSAINT (Table 6) and Cluster-GCN (Table 7). For the baseline, we use the same hyperparameters reported in the corresponding paper. We make two main observations. First, INT2 quantization works under a much smaller batch size. Namely, for both Cluster-GCN and GraphSAINT, the accuracy drop is negligible even when the sampled subgraph contains only $\approx 500$ nodes. **This observation implies that the activations of GNNs can be aggressively compressed regardless of the batch size.** Second, using a much smaller batch size in general will lead to an accuracy drop. However, for Yelp, a much smaller batch size may even improve the performance.

Table 6: The ablation study of the effect of batch size to GraphSAINT. Here "Small BS" means smaller batch size. For GraphSAINT, INT2 quantization also works under the smaller batch size.

| Dataset | Method | Walk Length | Root | Batch Size | Accuracy (%) / F1-micro |
|---|---|---|---|---|---|
| Flickr | Baseline | 2 | 6000 | 12,000 | 51.11 ± 0.28 |
| | Small BS | 1 | 500 | 500 | 47.67 ± 0.42 |
| | Small BS w./ INT2 quantization | 1 | 500 | 500 | 48.33 ± 0.29 |
| Reddit | Baseline | 4 | 2000 | 8,000 | 96.02 ± 0.08 |
| | Small BS | 1 | 500 | 500 | 93.73 ± 0.14 |
| | Small BS w./ INT2 quantization | 1 | 500 | 500 | 93.72 ± 0.10 |
| Yelp | Baseline | 2 | 1250 | 2,500 | 63.78 ± 0.12 |
| | Small BS | 1 | 500 | 500 | 64.05 ± 0.14 |
| | Small BS w./INT2 quantization | 1 | 500 | 500 | 64.01 ± 0.12 |

Table 7: The ablation study of the effect of batch size to Cluster-GCN. Here "Small BS" means smaller batch size. For Cluster-GCN, INT2 quantization also works under the smaller batch size.

| Dataset | Method | #partitions | #clusters per batch | Batch Size | Accuracy (%) / F1-micro |
|---|---|---|---|---|---|
| Flickr | Baseline | 1000 | 30 | 2,680 | 49.61 ± 0.47 |
| | Small BS | 1000 | 5 | 446 | 50.01 ± 0.30 |
| | Small BS w./ INT2 quantization | 1000 | 5 | 446 | 49.99 ± 0.28 |
| Reddit | Baseline | 1500 | 20 | 3,100 | 95.62±0.10 |
| | Small BS | 1500 | 3 | 465 | 95.30 ± 0.13 |
| | Small BS w./ INT2 quantization | 1500 | 3 | 465 | 95.29 ± 0.07 |
| Yelp | Baseline | 5000 | 20 | 2,870 | 63.98 ± 0.14 |
| | Small BS | 5000 | 3 | 430 | 63.91 ± 0.14 |
| | Small BS w./INT2 quantization | 5000 | 3 | 430 | 63.69 ± 0.23 |

Table 8: The Operation configurations of EXACT.

| Operations | Quantization? | Random Projection? | Extra Errors? |
|---|---|---|---|
| Linear (MM) | ✓ | ✓ | ✓ |
| SPMM | ✓ | ✓ | ✓ |
| SPMM_MEAN | ✓ | ✓ | ✓ |
| SPMM_MAX | ✓ | ✓ | ✓ |
| SPMM_MIN | ✓ | ✓ | ✓ |
| BatchNorm | ✓ | ✗ | ✓ |
| ReLU | ✓(fixed 1 bit) | ✗ | ✗ |
| Dropout | ✓(fixed 1 bit) | ✗ | ✗ |

# F    SYSTEM IMPLEMENTATION OF EXACT

## F.1    INDIVIDUAL LAYERS CONFIGURATIONS OF EXACT

The operation configurations are shown in Table 8. For all graph convolution operations, EXACT can apply both the quantization and random projection to their saved activation maps. In EXACT, the SPMM and its variants (i.e., SPMM_MAX, SPMM_MEAN, and SPMM_MIN) are implemented based

on those provided in Pytorch Sparse[3], with extra supporting for compressing activation maps by quantization and random projection.

For BatchNorm layers, we found that quantizating its saved activation maps only impact model performance a little, which is experimentally verified in the experiments. However, we experimentally found that randomly projecting its saved activation maps will lead to divergence. We note that this observation is consistent with previous finding that BatchNorm is very sensitive to noise (Micikevicius et al., 2017). Hence, EXACT only apply the quantization to BatchNorm layers.

Regarding ReLU operations, we have $y = \text{ReLU}(x) = x\mathbf{1}_{x>0}$ and $\nabla y = \nabla y \mathbf{1}_{x>0}$. Hence, ReLU operations only need to store $\mathbf{1}_{x>0}$ in the context for the backward pass, which takes a single bit per element to store, without introducing any errors. Since standard deep learning framework only support down to INT8 precision, here we convert the mask matrix into bit-stream and fuse this process into the CUDA kernel of ReLU.forward to minimize the time overhead.

Regarding Dropout operations, let $p$ be the dropout probability. During the training process, we have

$$y = \text{Dropout}(x, p) = \frac{1}{1-p}xg, \qquad \nabla x = \frac{1}{1-p}\nabla y g, \tag{15}$$

where $g \sim \text{Bernoulli}(1-p)$ is a binary vector sharing the same shape as $x$. $\frac{1}{1-p}$ is the normalization factor such that $\mathbb{E}[y] = x$. Similarly, Dropout Operations also only need to store $g$ in the context for the backward pass, which takes a single bit per element to store, without introducing any errors. Similarly, we convert $g$ into bit-stream and fuse this process into the CUDA kernel of Dropout.forward to minimize the time overhead..

To be clear, "fixed 1 bit" in Table 8 means the activation maps of ReLU and Dropout operations take only 1 bit per element to store. And the bit-width of all other operations can be adjusted.

### F.2 IMPLEMENTATION DETAILS

We provide simple API for converting modules in Pytorch Geometric and Pytorch to its corresponding version in EXACT. For example, replacing `torch_geometric.nn.GCNConv` with `exact.GCNConv` and replacing `torch.dropout` with `exact.dropout`. Currently, EXACT only support Pytorch Geometric and Pytorch. In future we will try to integrate EXACT with other popular graph learning packages, such as DGL.

Following ActNN (Chen et al., 2021a), to obtain the highest compression ratio, the quantization range $r_v^{(l)}$ and the zero point $Z_v^{(l)}$ are stored in the bfloat16 data type [4]. The quantization and dequantization modules are both implemented using `CUDA` kernels. We note that Pytorch only support data types down to INT8. To obtain highest compression ratio, the quantized data is compressed into bit streams such that it can be decoded later during the dequantization process.

All CUDA kernels in EXACT support both full-precision and half-precision (i.e., bfloat16 and float16). Thus, EXACT can also be integrated with the automated mixed precision training, i.e., AMP[5], to further decrease the memory consumption.

## G EXPERIMENT SETTINGS

### G.1 DATASETS, FRAMEWORKS, AND HARDWARES

We give the detailed statistics and the download URLs for all datasets used in our experiments in Table 10. We follow the standard data splits and all datasets are directly downloaded from Pytorch Geometric or the protocol of OGB (Hu et al., 2020). We implement all models based on Pytorch and Pytorch Geometric. Almost all experiments are done on a single NVIDIA GeForce RTX 3090 with 24GB GPU memory. During our experiments, we found that **the version of Pytorch, Pytorch Sparse, and Pytorch Scatter can significantly impact the running speed of the baseline.** Here we list the details of our used packages in all experiments in Table 9.

---

[3]https://github.com/rusty1s/pytorch_sparse/blob/master/torch_sparse/matmul.py
[4]https://pytorch.org/docs/stable/generated/torch.Tensor.bfloat16.html
[5]https://pytorch.org/docs/stable/amp.html

Table 9: Package configurations of our experiments.

| Package | Version |
|---|---|
| CUDA | 11.1 |
| pytorch_sparse | 0.6.12 |
| pytorch_scatter | 2.0.8 |
| pytorch_geometric | 1.7.2 |
| pytorch | 1.9.0 |
| OGB | 1.3.1 |

Table 10: Dataset Statistics.

| Dataset | Task | Nodes | Edges | Features | Classes | Label Rates |
|---|---|---|---|---|---|---|
| Reddit[6] | multi-class | 232,965 | 11,606,919 | 602 | 41 | 65.86% |
| Flickr [7] | multi-class | 89,250 | 449,878 | 500 | 7 | 50.00% |
| Yelp [8] | multi-label | 716,847 | 6,977,409 | 300 | 100 | 75.00% |
| *ogbn-arxiv* [9] | multi-class | 169,343 | 1,157,799 | 128 | 40 | 53.70% |
| *ogbn-products* [10] | multi-class | 2,449,029 | 61,859,076 | 100 | 47 | 8.03% |

## G.2 MODEL HYPERPARAMETER CONFIGURATIONS OF TABLE 1 AND TABLE 2

Table 11: Training configuration of Full-Batch GCN, GraphSAGE, and GAT in Table 1 and Table 2.

| Model | Training | | | | Architecture | | | |
|---|---|---|---|---|---|---|---|---|
| | Learning Rates | Epochs | Dropout | Gradient Clipping | BatchNorm | Layers | Hidden Dimension | Heads |
| GCN | 0.01 | 500 | 0.5 | 0.0 | Yes | 3 | 128 | - |
| GraphSAGE | 0.01 | 500 | 0.5 | 0.0 | Yes | 3 | 128 | - |
| GAT | 0.002 | 2000. | 0.75 | 0.0 | Yes | 3 | 128 | 3 |

We adopt three popular GNNs. Namely, GCN (Kipf & Welling, 2017), GraphSAGE (Hamilton et al., 2017), and GAT (Veličković et al., 2017) in Table 1 and Table 2. We follow the hyperparameter configurations and codebases provided on the OGB (Hu et al., 2020) leader-board. Specifically, the hyperparameter configuration is given in Table 11. The optimizer is Adam (Kingma & Ba, 2014) for GCN and GraphSAGE, while the optimizer is RMSprop for GAT. All methods terminate after a fixed number of epochs. We report the test accuracy associated with the highest validation score. The "Gradient Clipping" in Table 11 indicate the maximum norm for gradients. "Gradient Clipping= 0.0" means we do not clip the gradients in that experiment.

## G.3 HYPERPARAMETER CONFIGURATIONS OF EXACT (RP+INT2)

The EXACT (RP+INT2) has only one hyperparameter, namely, $\frac{D}{R}$. The $\frac{D}{R}$ configuration of EXACT (RP+INT2) can be found in Table 12.

## G.4 MODEL HYPERPARAMETER CONFIGURATIONS IN SECTION 5

The optimizer used in all experiments in Section 5 is Adam (Kingma & Ba, 2014). We use the default hyperparameters for Adam optimizer, except for the learning rate. All methods terminate after a fixed number of epochs. We report the test accuracy/F1-micro associated with the highest validation score. Regarding Reddit, Flickr, and Yelp dataset, we follow the hyperparameter configurations reported in the respective papers as closely as possible. We clips the gradient during training. The "Gradient Clipping" in below tables indicate the maximum norm for gradients. "Gradient Clipping= 0.0" means we do not clip the gradients in that experiment.

---

[6]https://pytorch-geometric.readthedocs.io/en/latest/modules/datasets.html#torch_geometric.datasets.Reddit

[7]https://pytorch-geometric.readthedocs.io/en/latest/modules/datasets.html#torch_geometric.datasets.Flickr

[8]https://pytorch-geometric.readthedocs.io/en/latest/modules/datasets.html#torch_geometric.datasets.Yelp

[9]https://ogb.stanford.edu/docs/nodeprop/#ogbn-arxiv

[10]https://ogb.stanford.edu/docs/nodeprop/#ogbn-products

Table 12: The $\frac{D}{R}$ configuration of EXACT (RP+INT2) in Table 3

| | Reddit | Flickr | Yelp | *ogbn-arxiv* | *ogbn-*products |
|---|---|---|---|---|---|
| Cluster-GCN | 8 | 8 | 4 | - | 2 |
| GraphSAINT | 8 | 8 | 8 | 8 | 2 |
| GCN | 8 | 8 | 8 | 8 | - |
| GraphSAGE | 8 | 8 | 4 | 8 | 4 |
| GCNII | 8 | 8 | 2 | 8 | - |

Regarding *ogbn-arxiv* and *ogbn-products* dataset, we follow the hyperparameter configurations and codebases provided on the OGB (Hu et al., 2020) leader-board. Please refer to the OGB website for more details. Table 13 and Table 14 summarize the hyperparameter configuration of Cluster-GCN and GraphSAINT, respectively. Table 15, Table 16, and Table 17 summarize the hyperparameter configuration of full-Batch GCN, full-Batch GraphSAGE, and full-batch GCNII, respectively.

Table 13: Training configuration of Cluster-GCN in Table 3.

| Dataset | Cluster Sampler | | Training | | | | Archtecture | | |
|---|---|---|---|---|---|---|---|---|---|
| | #partitions | #Cluster per batch | Learning Rates | Epochs | Dropout | Gradient Clipping | BatchNorm | Layers | Hidden Dimension |
| Reddit | 1500 | 20 | 0.01 | 40 | 0.1 | 0.5 | Yes | 2 | 128 |
| Flickr | 1000 | 30 | 0.01 | 15 | 0.2 | 0.5 | Yes | 2 | 256 |
| Yelp | 5000 | 20 | 0.01 | 75 | 0.1 | 0.5 | Yes | 2 | 512 |
| *ogbn-products* | 15000 | 32 | 0.001 | 50 | 0.5 | 0.0 | No | 3 | 256 |

Table 14: Training configuration of GraphSAINT in Table 3.

| Dataset | RandomWalk Sampler | | Training | | | | Archtecture | | |
|---|---|---|---|---|---|---|---|---|---|
| | Walk length | Roots | Learning Rates | Epochs | Dropout | Gradient Clipping | BatchNorm | Layers | Hidden Dimension |
| Reddit | 4 | 2000 | 0.01 | 40 | 0.1 | 0.5 | Yes | 2 | 128 |
| Flickr | 2 | 6000 | 0.01 | 15 | 0.2 | 0.5 | Yes | 2 | 256 |
| Yelp | 2 | 1250 | 0.01 | 75 | 0.1 | 0.5 | Yes | 2 | 512 |
| *ogbn-arxiv* | 3 | 10000 | 0.01 | 500 | 0.5 | 0.5 | Yes | 3 | 256 |
| *ogbn-products* | 3 | 20000 | 0.01 | 20 | 0.5 | 0.0 | No | 3 | 256 |

Table 15: Training configuration of Full-Batch GCN in Table 3.

| Dataset | Training | | | | Archtecture | | |
|---|---|---|---|---|---|---|---|
| | Learning Rates | Epochs | Dropout | Gradient Clipping | BatchNorm | Layers | Hidden Dimension |
| Reddit | 0.01 | 400 | 0.5 | 0.5 | Yes | 2 | 256 |
| Flickr | 0.01 | 400 | 0.3 | 0.5 | Yes | 2 | 256 |
| Yelp | 0.01 | 500 | 0.1 | 0.5 | Yes | 2 | 512 |
| *ogbn-arxiv* | 0.01 | 500 | 0.5 | 0.5 | Yes | 3 | 128 |

Table 16: Training configuration of Full-Batch GraphSAGE in Table 3.

| Dataset | Training | | | | Archtecture | | |
|---|---|---|---|---|---|---|---|
| | Learning Rates | Epochs | Dropout | Gradient Clipping | BatchNorm | Layers | Hidden Dimension |
| Reddit | 0.01 | 400 | 0.5 | 0.5 | Yes | 2 | 256 |
| Flickr | 0.01 | 400 | 0.3 | 0.5 | Yes | 2 | 256 |
| Yelp | 0.01 | 500 | 0.1 | 0.5 | Yes | 2 | 512 |
| *ogbn-arxiv* | 0.01 | 500 | 0.5 | 0.5 | Yes | 3 | 128 |
| *ogbn-products* | 0.002 | 500 | 0.5 | 0.5 | No | 3 | 256 |

Table 17: Training configuration of Full-Batch GCNII in Table 3.

| Dataset | Training | | | | Archtecture | | |
|---|---|---|---|---|---|---|---|
| | Learning Rates | Epochs | Dropout | Gradient Clipping | BatchNorm | Layers | Hidden Dimension |
| Reddit | 0.01 | 400 | 0.5 | 0.5 | Yes | 4 | 256 |
| Flickr | 0.01 | 400 | 0.5 | 0.5 | Yes | 8 | 256 |
| Yelp | 0.01 | 500 | 0.1 | 0.5 | Yes | 4 | 512 |
| *ogbn-arxiv* | 0.001 | 1000 | 0.1 | 0.1 | Yes | 16 | 256 |

# H   ANALYZING THE MEMORY USAGE

## H.1   ANALYZING THE MEMORY USAGE

We use `torch.cuda.memory_allocated` for the memory measurement. As we mentioned in Table 3, "Act Mem." is the memory occupied by activation maps. Besides activation maps, the model, optimizer, input data, weight gradients, and activation gradients also occupy GPU memory. We will analyze each of them below. **First**, the memory occupied by the model, optimizer, and weight gradients is negligible because the number of parameters in most of GNNs is very small. **Second**, the memory occupied by the input data depends on the graph size and often cannot be compressed. This part can take up a lot of memory when the graph is large. **Third**, the memory occupied by activation gradients is dynamic and hard to estimate, since these tensors are temporarily stored in GPUs. For standard deep learning library, they will be deleted as soon as their reference counting becomes zero (Abadi et al., 2016; Paszke et al., 2019). EXACT cannot compress activation gradients. However, the memory occupied by activation gradients can be compressed using AMP, because activation gradients are stored in float16 data type under AMP. As illustrated in Appendix F.2, we note that our EXACT framework can be integrated with AMP.

The memory usage of the model weights plus optimizer is about 2MB. Since activation gradients are dynamic and hard to analyze, here we report the peak memory usage during the backward pass, which encompass the activation gradients and other intermediate variables. For the memory usage of the input data, activation maps, and the peak memory usage during the backward pass, we provided a detailed analysis in Table 18. We can observe that the memory usage is mainly occupied by activation maps. For GCNII, the memory usage is dominated by the activation maps. This is because we need to store all layers' activation maps during the forward pass. In contrast, during the backward pass, there is usually only one layer' activation gradients are kept in memory.

Table 18: The detailed analysis about the memory usage of input data, activation maps, and peak memory usage during the backward pass. "Data Mem" is the memory usage of input data (including the input feature matrix $X$, adjacency matrices $A$, and labels )."Act Mem" is the memory usage of activation maps. "Peak BWD Mem" is the peak memory usage during the backward pass. "Ratio (%)" here equals $\frac{\text{Act Mem}}{\text{Data Mem}+\text{Act Mem}+\text{Peak BWD Mem}}$.

| | Reddit | | | | Flickr | | | | Yelp | | | | ogbn-arxiv | | | | ogbn-products | | | |
|---|---|---|---|---|---|---|---|---|---|---|---|---|---|---|---|---|---|---|---|---|
| | Data Mem | Act Mem. | Peak BWD Mem. | Ratio (%) | Data Mem | Act Mem | Peak BWD Mem. | Ratio (%) | Data Mem | Act Mem | Peak BWD Mem. | Ratio (%) | Data Mem | Act Mem | Peak BWD Mem. | Ratio (%) | Data Mem | Act Mem | Peak BWD Mem. | Ratio (%) |
| Cluster-GCN | 8.8 | 15 | 3 | 60.0 | 5.5 | 16.5 | 5.3 | 60.4 | 5.3 | 29.3 | 12 | 62.9 | - | - | - | - | 4.9 | 35.2 | 11.5 | 68.2 |
| GraphSAINT | 27.6 | 44 | 9.3 | 54.4 | 31.2 | 88.7 | 29 | 59.6 | 5.8 | 33.5 | 13 | 64.0 | 26.7 | 270 | 62.4 | 75.1 | 57 | 516 | 157 | 70.7 |
| GCN | 1168 | 1029 | 316 | 40.9 | 208 | 379 | 86 | 56.3 | 1544 | 6429 | 1195 | 70.1 | 175.7 | 729.4 | 82.2 | 73.9 | - | - | - | - |
| GraphSAGE | 1168 | 1527 | 696 | 45.0 | 208 | 547 | 184 | 58.3 | 1544 | 6976 | 2881 | 61.2 | 175.7 | 786.2 | 192 | 68.1 | 4811 | 16555 | 6176 | 60.1 |
| GCNII | 1168 | 5850 | 239 | 80.6 | 208 | 4067 | 88 | 93.2 | 1544 | 33540 | 1197 | 92.4 | 175.7 | 14409 | 194 | 97.5 | - | - | - | - |

## H.2   OVERALL COMPRESSION RATIO

The overall memory compression ratio is shown in Table 19. We observe that (1) for shallow GNN models, the overall memory compression ratio ranges from $1.5\times$ to $4\times$. (2) for GCNII, the overall memory compression ratio ranges from $4\times$ to $18\times$. We note that we store the graph structure data (e.g., node ID and edge ID) in torch.Long data type, which can safely cast to torch.Int data type to save the memory. Our implementation supports for using torch.Int as the data type for the graph structure data. However, for a fair comparison, we do not utilize this feature.

Table 19: The detailed analysis for the overall memory compression ratio. Below the equation means "Data Mem" + "Act Mem" + "Peak BWD Mem" = "Overall Mem". EXACT can only compress the memory usage of activation maps.

| Model | Method | Reddit | Flickr | Yelp | ogbn-arxiv | ogbn-products |
|---|---|---|---|---|---|---|
| Cluster-GCN | Baseline | 8.8+14.5+3=26.8 | 5.5+16.5+5.3=27.3 | 5.3+29.3+12=46.6 | - | 4.9+35.2+11.5=51.6 |
| | EXACT (INT2) | 8.8+2+3=13.8 (1.94×) | 5.5+1.5+5.3=12.3 (2.22×) | 5.3+4+12=21.3 (2.19×) | - | 4.9+2.5+11.5=18.9 (2.73×) |
| | EXACT (RP+INT2) | 8.8+1.4+3=13.2 (2.03×) | 5.5+0.9+5.3=11.7 (2.33×) | 5.3+3+12=20.3 (2.30×) | | 4.9+2.2+11.5=18.6 (2.77×) |
| Graph-Saint | Baseline | 27.6+44.3+9.3=81.2 | 31.2+88.7+29=148.9 | 5.8+33.5+13=52.3 | 26.7+270+62.4=359.1 | 57+516+157=730 |
| | EXACT (INT2) | 27.6+6.6+9.3=43.5 (1.87×) | 31.2+7.8+29=68 (2.19×) | 5.8+4.3+13=23.1 (2.26×) | 26.7+20+62.4=109.1 (3.29×) | 57+40.5+157=254.5 (2.87×) |
| | EXACT (RP+INT2) | 27.6+3.6+9.3=40.5 (2.00×) | 31.2+3.4+29=63.6 (2.34×) | 5.8+3.3+13=22.1 (2.37×) | 26.7+10.8+62.4=99.9 (3.59×) | 57+29.5+157=243.5 (3.00×) |
| GCN | Baseline | 1168+1029+316=2513 | 208+378.8+86=672.8 | 1544+6429+1195=9168 | 175.7+729.4+82.2=987.3 | - |
| | EXACT (INT2) | 1168+122.8+316=1607 (1.56×) | 208+37+86=331 (2.03×) | 1544+640+1195=3379 (2.71×) | 175.7+54.5+82.2=312.4 (3.16×) | - |
| | EXACT (RP+INT2) | 1168+67+316=1551 (1.62×) | 208+17.8+86=311.8 (2.16×) | 1544+427+1195=3166 (2.90×) | 175.7+30.2+82.2=288.1 (3.43×) | - |
| Graph-SAGE | Baseline | 1168+1527+696=3391 | 208+547+184=939 | 1544+6976+2881=11401 | 175.7+786.2+192=1153.9 | 4811+16555+6176=27542 |
| | EXACT (INT2) | 1168+156+696=2020 (1.68×) | 208+49.3+184=441.3 (2.13×) | 1544+680+2881=5105 (2.23×) | 175.7+60.8+192=428.5 (2.69×) | 4811+1144+6176=12131 (2.27×) |
| | EXACT (RP+INT2) | 1168+72+696=1936 (1.75×) | 208+20.4+184=412.4 (2.28×) | 1544+466.5+2881=4891.5 (4.33×) | 175.7+30.5+192=398.2 (2.90×) | 4811+572+6176=11559 (2.38×) |
| GCNII | Baseline | 1168+5850+239=7257 | 208+4067+88=4363 | 1544+33540+1197=36281 | 175.7+14409+194=14778.7 | - |
| | EXACT (INT2) | 1168+388+239=1795 (4.04×) | 208+256.4+88=552.4 (7.89×) | 1544+2236+1197=4977 (7.29×) | 175.7+899+194=1268.7 (11.65×) | - |
| | EXACT (RP+INT2) | 1168+198+239=1605 (4.52×) | 208+127.6+88=423.6 (10.30×) | 1544+1649+1197=4390 (8.26×) | 175.7+451.2+194=820.9 (18.00×) | - |

## H.3 ANALYZING THE MEMORY COMPRESSION RATIO

Below we analyze the compress ratio in Table 3. We take a three-layer, 128-dimensional GCNs trained on *ogbn-arxiv* for example. The computational graph of the baseline in Table 3 is:

$$
\begin{aligned}
\text{Total bits} =\ & 0(\texttt{MM}) + 32(\texttt{SPMM}) + 32(\texttt{BN}) + 0(\text{ReLU}) + 8(\text{Dropout}) + && \text{(the first layer)} \\
& 32(\texttt{MM}) + 32(\texttt{SPMM}) + 32(\texttt{BN}) + 0(\text{ReLU}) + 8(\text{Dropout}) + && \text{(the second layer)} \\
& 32(\texttt{MM}) + 32(\texttt{SPMM}) && \text{(the third layer)}
\end{aligned}
$$

Hence the baseline costs totally 240 bit per element. The first MM in the first layer does not cost extra bits because its activation map is exactly the input feature matrix $X$, which has been stored in GPU memory (recall that we need to first move the input data to GPU memory before training). Pytorch will save this part of memory via the "pass by reference" mechanism. The official ReLU operation in Pytorch does not need the extra space for saving activation maps (they can reuse the activation maps saved by the previous layer via passing by reference). Regarding Dropout operation, Pytorch stores the mask matrix using UINT8 data type, which costs 8 bit per element.

The computational graph of "EXACT (INT2)" in Table 3 is:

$$
\begin{aligned}
\text{Total bits} =\ & 2.25(\texttt{MM}) + 2.25(\texttt{SPMM}) + 2.25(\texttt{BN}) + 1(\text{ReLU}) + 1(\text{Dropout}) + && \text{(the first layer)} \\
& 2.25(\texttt{MM}) + 2.25(\texttt{SPMM}) + 2.25(\texttt{BN}) + 1(\text{ReLU}) + 1(\text{Dropout}) + && \text{(the second layer)} \\
& 2.25(\texttt{MM}) + 2.25(\texttt{SPMM}) && \text{(the third layer)}
\end{aligned}
$$

The 2.25 bit of MM, SPMM, and BN is from 2 (quantized activation maps) + 0.125 (the zero point tensor) + 0.125 (the range tensor), where $0.125 = \frac{16 \text{ (the bit-width of bfloat16)}}{128 \text{ (the dimension is 128)}}$ (See Appendix F.2 for details). Also for EXACT, we cannot leverage the "pass by reference" mechanism to save memory for ReLU since the exact activation maps of previous operation is replaced by the compressed one. Hence in EXACT, the 1-bit masks of ReLU and Dropout is converted into bit-streams using CUDA kernels, which cost 1 bit per element to store.

Hence the "EXACT (INT2)" costs totally 22 bit per element. And the theoretical compression ratio is $\frac{240}{22} = 10.9$. The empirical compression ratio in Table 3 may have a small gap with the theoretical one. This is because the attributes of the sparse tensor (adjacency matrix) in pytorch_sparse is lazily initialized, i.e., it may be generated during the forward pass and be account for the memory of activation maps.

The computational graph of "EXACT (RP+INT2)" with $\frac{D}{R} = 8$ (see Table 12) in Table 3 is:

$$
\begin{aligned}
\text{Total bits} =\ & 0.28(\texttt{MM}) + 0.28(\texttt{SPMM}) + 2.25(\texttt{BN}) + 1(\text{ReLU}) + 1(\text{Dropout}) + && \text{(the first layer)} \\
& 0.28(\texttt{MM}) + 0.28(\texttt{SPMM}) + 2.25(\texttt{BN}) + 1(\text{ReLU}) + 1(\text{Dropout}) + && \text{(the second layer)} \\
& 0.28(\texttt{MM}) + 0.28(\texttt{SPMM}) && \text{(the third layer)}
\end{aligned}
$$

The 0.28 bit of MM and SPMM is from $\frac{2}{8} + \frac{16}{8 \times 128} + \frac{16}{8 \times 128} = 0.28125$. We note that EXACT does not apply random projection for BatchNorm layers (see Appendix F.2). Hence the "EXACT

(RP+INT2)" costs totally 10.18 bit per element. And the theoretical compression ratio is $\frac{240}{10.18} = 23.57$. Again, there exists gap between the empirical compression ratio and the theoretical one because the lazy initialization mechanism, the existence of temporary tensors, and the neglect of the memory usage of these random projection matrices.

We emphasize that **the compression ratio also depends on the number of input features of the dataset.** (because the mentioned "input feature matrix $X$ is passed by reference" mechanism). For the above example, the number of input features and the hidden dimension are both 128 and hence it is easy to be analyzed. Hence, the theoretical compression ratio may vary for different datasets.

# I  ADDITIONAL EXPERIMENT RESULTS

## I.1  MORE RESULTS ON ACCURACY AGAINST THE PRECISION AND $\frac{D}{R}$

To support the claim that "the performance of EXACT is mainly determined by the $\frac{D}{R}$ ratio of random projection", we present more results on the test accuracy against the precision and $\frac{D}{R}$ ratio here. In Figure 5, we show the results of two models trained with EXACT using full-batch data on the *ogbn-arxiv* dataset. In Figure 6, we show the results of two models trained with EXACT using mini-batch data on Yelp dataset.

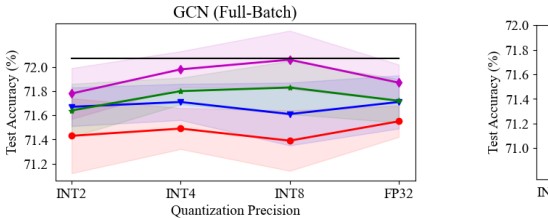

Figure 5: The performance of EXACT is mainly determined by the $\frac{D}{R}$ ratio of random projection. The dataset here is *ogbn-arxiv*. All reported results are averaged over ten random trials.

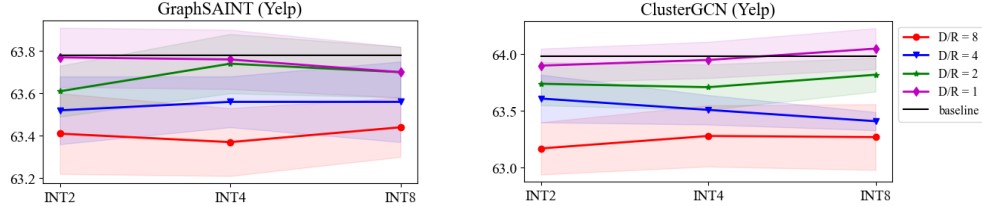

Figure 6: The performance of EXACT is mainly determined by the $\frac{D}{R}$ ratio of random projection. The dataset here is Yelp. All reported results are averaged over ten random trials.

## I.2  COMPARISON AGAINST SWAPPING AND GRADIENT CHECKPOINTING

We also compare EXACT against the naive swapping and the naive gradient checkpointing. Both swapping and gradient checkpointing are lossless compression, so they do not have any accuracy drop. Below we present and discuss about their trade-off among the space and speed.

For swapping, we simply offload all activation maps to CPU memory. Thus, it can achieve the highest compression ratio for activation maps. However, its running time overhead is roughly 50%-80%, which is often unacceptable.

For gradient checkpointing, we utilize `torch.utils.checkpoint` to insert checkpoints at each GNN layer. For the time overhead, as shown in Figure 3, it is comparable to EXACT. We present the memory usage of activations in Table 20. In summary, its time overhead is comparable to EXACT, however, the memory compression ratio of activations is $1.7 \sim 2.3\times$, which is not large enough. Thus, we still cannot train the GCNII on Yelp dataset using a RTX 3090 (24GB) GPU.

Table 20: The memory usage (MB) of activation maps with the gradient checkpointing. "OOM" means out-of-memory. In general, the compression ratio of activation maps are $1.7 \sim 2.3\times$.

| Model | ogbn-arxiv | Yelp | Reddit |
|---|---|---|---|
| GCN | 425 | 4493 | 722 |
| GraphSAGE | 424 | 3913 | 1030 |
| GCNII | 6236 | OOM | 3366 |

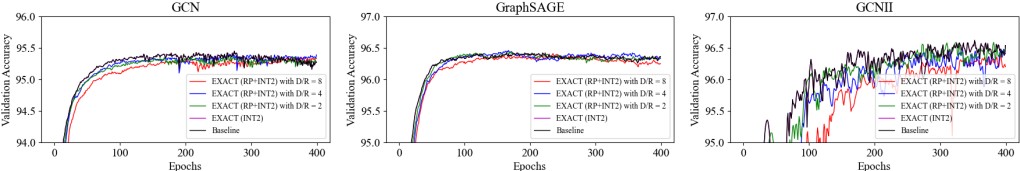

Figure 7: Validation Accuracy on Reddit dataset using EXACT with different configurations.

### I.3 THE TRAINING CURVES OF EXACT

From the optimization theory, common convergence speed bounds (measured by the number of iterations) in stochastic optimization improve with smaller gradient variance (Bottou et al., 2018). EXACT essentially trades the gradient variance in return for reduced memory. Here we experimentally examine how EXACT affects the convergence speed. Figure 7 shows the training curves of GNNs trained with EXACT using different configurations on Reddit dataset. We make two main observations. First, EXACT with smaller $\frac{D}{R}$ typically converges faster per iteration. This is consistent with the mentioned optimization theory. Second, when increasing the $\frac{D}{R}$ ratio, the difference in the convergence speed is very small, where the convergence speed is measured by the gap in validation accuracy between consecutive epochs. In practical scenarios, EXACT hence only have limited impact on the model performance.

### I.4 HYPERPARAMETER SENSITIVITY EXPERIMENT

As we analyzed in the main body, INT2 is a suitable precision and we only vary the $\frac{D}{R}$ ratio for EXACT. In this section, we investigate the sensitivity of EXACT (RP+INT2) to the $\frac{D}{R}$ ratio. Table 12 shows the configuration of EXACT (RP+INT2) in the experiment in the main body. Here we present a comprehensive hyperparameter sensitivity study for EXACT. Specifically, the sensitivity studies of EXACT with GraphSAINT, ClusterGCN, GCN, GraphSAGE, and GCNII are shown in Figure 8, Figure 9, Figure 10, Figure 11, and Figure 12, respectively. Here we summarize some key observations. First, the model performance drop generally increases with the $\frac{D}{R}$ ratio. Second, when $\frac{D}{R} = 8$, the loss in accuracy of EXACT (RP+INT2) is below $0.5\%$ on two third of experiments. To be concrete, in Table 3, we totally adopt 22 combinations of different datasets and models. As shown in Table 3 and Table 8, for 15 of the 22 experiments, the loss of accuracy is below or near $0.5\%$ when $\frac{D}{R} = 8$. Third, when $\frac{D}{R} = 8$, the loss in accuracy is below $1\%$ in almost all experiments, excepting for two experiments. Namely, ClusterGCN with *ogbn-products* and GraphSAINT with *ogbn-products*.

### I.5 COMPARISON BETWEEN EXACT AND SAMPLING METHODS

### I.5.1 COMPARE EXACT TO SAMPLING METHODS UNDER A FIXED MEMORY BUDGET

As examined in Table 3, EXACT and subgraph sampling methods are orthogonal and can be applied over each others. Here we present an ablation study of comparing EXACT to subgraph sampling methods in a standalone way. We note that EXACT only focus on saving the memory for activations. In contrast, subgraph sampling methods can simultaneously reduce the memory of the input data, activations, and activation gradients since they directly reduce the number of node embeddings

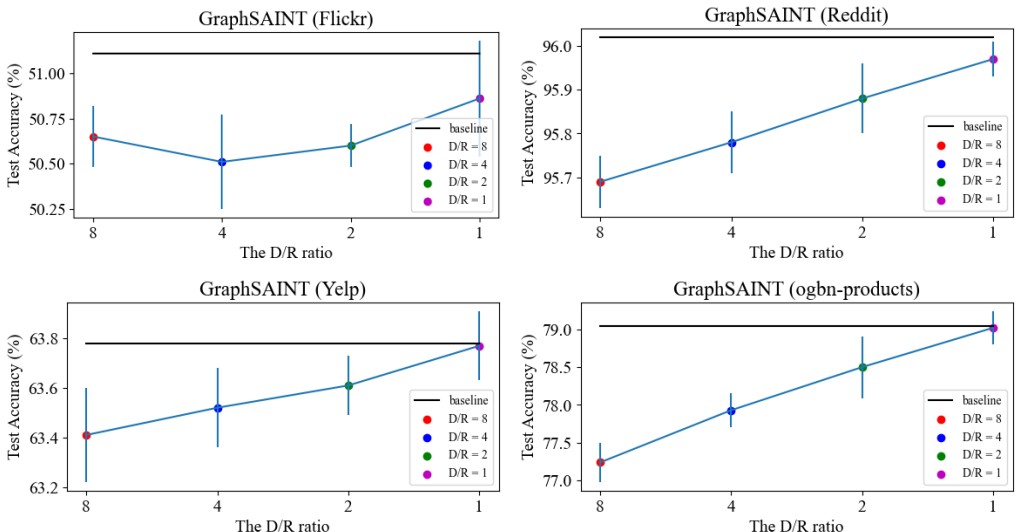

Figure 8: The sensitivity study of EXACT (RP+INT2) to the $\frac{D}{R}$ ratio, where the model is Graph-SAINT. All reported results are averaged over ten random trials.

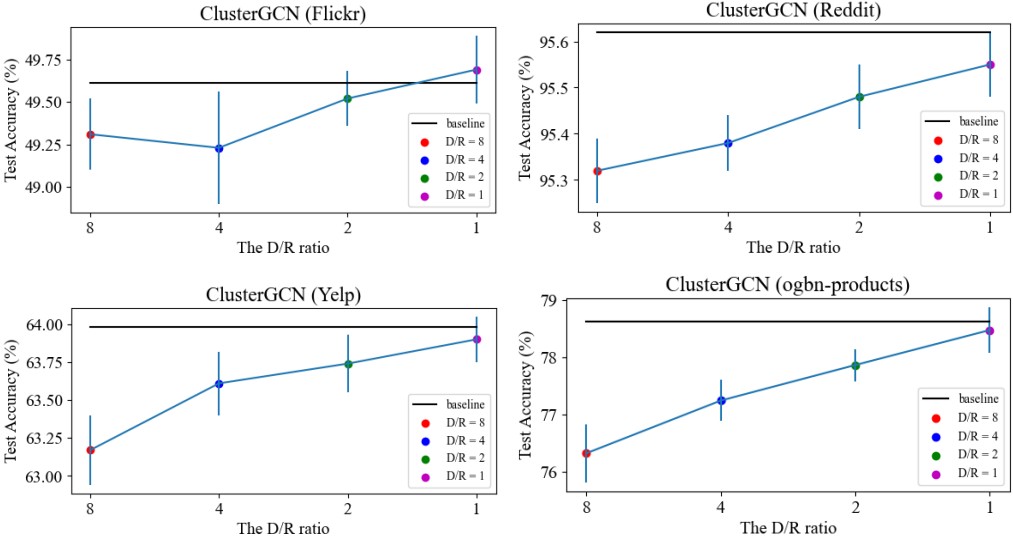

Figure 9: The sensitivity study of EXACT (RP+INT2) to the $\frac{D}{R}$ ratio, where the model is Cluster-GCN. All reported results are averaged over ten random trials.

retained in the memory. Hence compared to subgraph sampling methods, one limitation of EXACT is that the overall memory saving ratio of EXACT often cannot surpass that of sampling methods.

Under the full-batch setting, one way to further improve the overall memory saving ratio is to incorporate the graph compression methods with EXACT, such as the graph sparsification (Spielman & Srivastava, 2011) and graph coarsening (Cai et al., 2021a). However, it is beyond the scope of this paper. For a fair comparison, we control the batch size of subgraph subgraph sampling methods such that their activation memory usage equals to that of EXACT (INT2). For both EXACT(INT2) and EXACT(RP+INT2), we quote the "GraphSAGE" results from Table 3. For GraphSAINT and Cluster-GCN, we tune their batch size such that they have the same activation memory usage as EXACT (INT2). The results are shown in Table 21. We make two main observations. First, for Reddit and Flickr, full batch training with EXACT outperforms the two sampling methods. For Yelp, subgraph sampling methods are better than full batch training with EXACT. However, we note

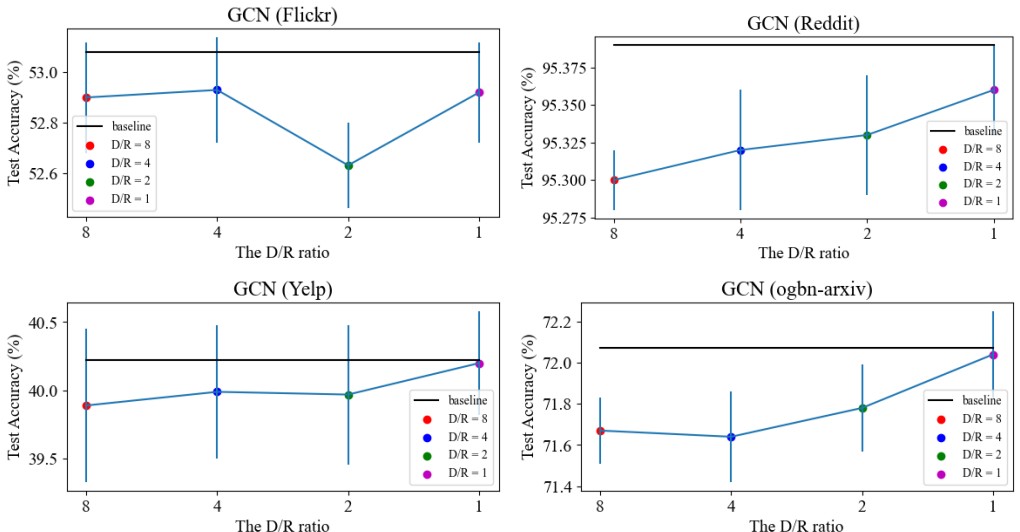

Figure 10: The sensitivity study of EXACT (RP+INT2) to the $\frac{D}{R}$ ratio, where the model is GCN. All reported results are averaged over ten random trials.

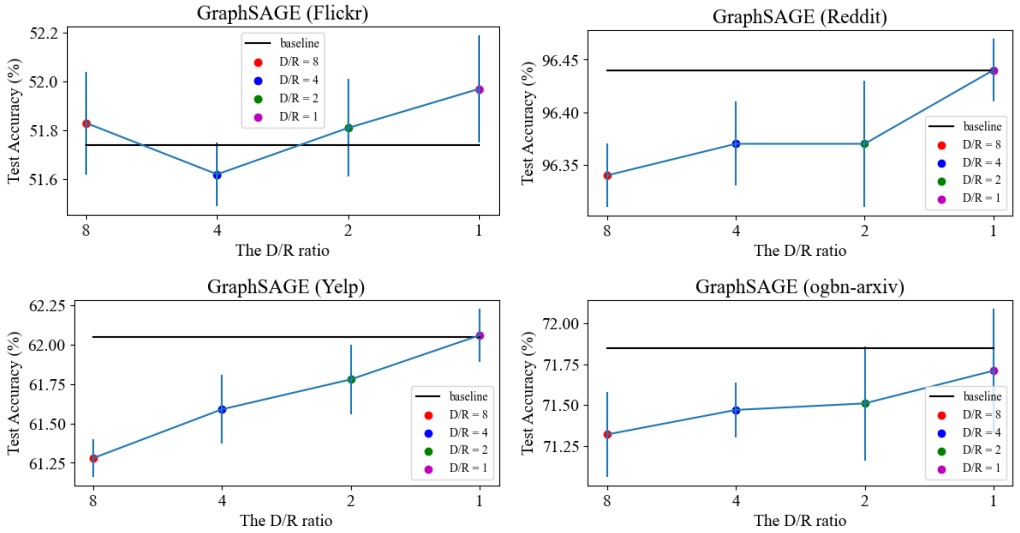

Figure 11: The sensitivity study of EXACT (RP+INT2) to the $\frac{D}{R}$ ratio, where the model is Graph-SAGE. All reported results are averaged over ten random trials.

that this gap is not from EXACT. As shown in Table 3, without EXACT, the F1-micro of full batch GraphSAGE is $62.05\%$, which is much lower than that of subgraph sampling methods. Also, from Table 6, one interesting observation is that for Yelp, a much smaller batch size ($\approx 500$) may even further improve the F1-micro of GraphSAINT from $63.20\%$ to $64.05\%$.

In summary, subgraph sampling methods may outperform the full batch training on some datasets (e.g., Yelp). In this case, we think full batch training with EXACT cannot outperform subgraph sampling methods. If this is not the case, as shown in Table 21, full batch training with EXACT can outperform subgraph sampling methods under a fixed activation memory usage.

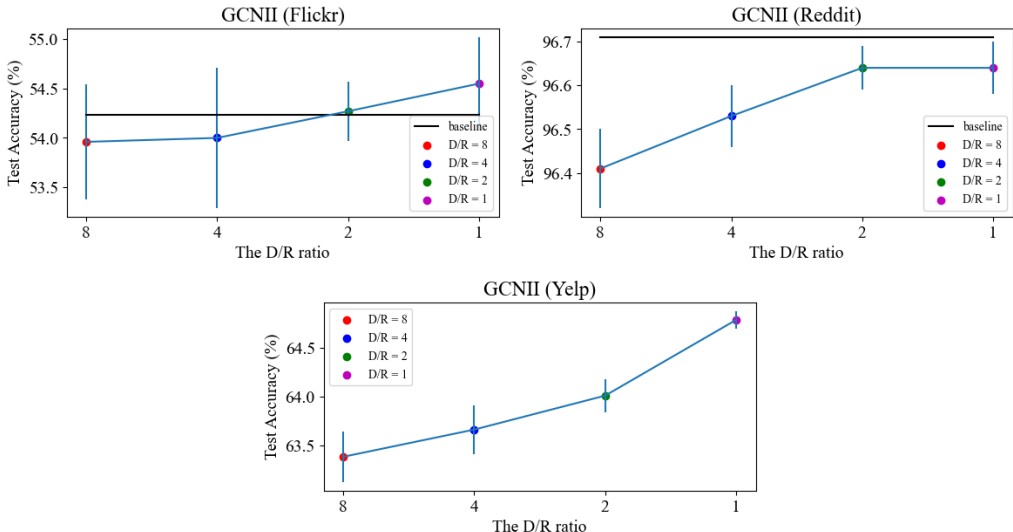

Figure 12: The sensitivity study of EXACT (RP+INT2) to the $\frac{D}{R}$ ratio, where the model is GCNII. All reported results are averaged over ten random trials.

Table 21: The ablation study of comparing GraphSAINT and Cluster-GCN to EXACT under a fixed memory budget for activations. For convenience, in this table, the activation memory usage of of full batch w./ EXACT (INT2), GraphSAINT, and Cluster-GCN are the same. And the activation memory usage of EXACT (RP+INT2) is lower than EXACT (INT2).

| Dataset | Method | Accuracy (%) / F1-micro |
|---|---|---|
| Flickr | GraphSAINT | 50.30 ± 0.16 |
| | Cluster-GCN | 49.98 ± 0.15 |
| | Full Batch w./ EXACT (INT2) | 51.97 ± 0.20 |
| | Full Batch w./ EXACT (RP+INT2) | 51.83 ± 0.21 |
| Reddit | GraphSAINT | 96.18 ± 0.03 |
| | Cluster-GCN | 96.03 ± 0.05 |
| | Full Batch w./ EXACT (INT2) | 96.40 ± 0.05 |
| | Full Batch w./ EXACT (RP+INT2) | 96.34 ± 0.03 |
| Yelp | GraphSAINT | 63.20 ± 0.19 |
| | Cluster-GCN | 63.26 ± 0.14 |
| | Full Batch w./ EXACT (INT2) | 61.95 ± 0.12 |
| | Full Batch w./ EXACT (RP+INT2) | 61.59 ± 0.12 |

### I.5.2 ENLARGING THE BATCH SIZE OF SAMPLING METHODS WITH EXACT

Here we present a case study of scaling up the batch size of GraphSAINT with EXACT on the *ogbn-products* dataset. As shown in Table 14, the batch size of GraphSAINT (number of nodes in the sampled subgraphs) is controlled by the "Walk length" and "Roots", and roughly equals "Walk length"×"Roots" (Zeng et al., 2020). For *ogbn-products*, we tripled the batch size by changing "Roots" from $20,000$ to $30,000$. All other hyperparameters are left unchanged. The results are shown in Table 22. We observe that EXACT may improve the accuracy when using larger batch size. We note that EXACT can scale up the batch size to more than $3\times$ larger. However, we found that if we further scale up the batch size to $4\times$, there is an accuracy drop compared to the baseline

Table 22: The test accuracy of GraphSAINT on the *ogbn-products*. All reported results are averaged over ten random trials.

| Method | Test Accuracy (%) |
|---|---|
| GraphSAINT | 79.03 ±0.23 |
| GraphSAINT + EXACT (INT2) w./ 3× batch size | 79.16 ±0.24 |
| GraphSAINT + EXACT (RP+INT2) w./ 3× batch size | 78.54 ±0.41 |

with the original batch size, regardless applying EXACT or not. This is consistent with previous finding that a larger batch size does not always lead to better performance for *ogbn-products* (Zeng et al., 2020; Hu et al., 2020). We quote the sentences from the OGB paper (Hu et al., 2020) to explain this counter-intuitive observation. "The recent mini-batch-based GNNs give promising results, even slightly outperforming the full-batch version of GraphSAGE that does not fit into ordinary GPU memory. The improved performance can be attributed to the regularization effects of mini-batch noise and edge dropout." (Hu et al., 2020).

In summary, in the *ogbn-products* ablation study, we utilize EXACT to triple the batch size of GraphSAINT, which may even improve the accuracy over the original one. Moreover, from our experiments, the 3× larger batch size performs the best on *ogbn-products*. Thus, for some datasets, there exist an optimal batch size for subgraph sampling methods. Since this optimal batch size may be beyond the capacity of the hardware, the meaning of EXACT is to enlarge the "search space" for finding this optimal batch size.

