# OpenReview forum: "EXACT: Scalable Graph Neural Networks Training via Extreme Activation Compression"
_ICLR.cc/2022/Conference — ICLR 2022 Poster_

### Official Review · Reviewer_eNSi · 2021-11-02

**Correctness:** 3
**Technical Novelty And Significance:** 3
**Empirical Novelty And Significance:** 3
**Recommendation:** 6
**Confidence:** 3

**Main Review:**

Limited memory capacity of GPUs impose severe constraints on the model/dataset size of GNNs that can be trained. That said, this paper tackles an important problem. On the positive side, the proposed technique is simple and based on proven ideas for tensor compression such as quantization and random projection. It was a pleasant surprise that those simple techniques work well to reduce the activation size with acceptable accuracy drops and computational overhead. Evaluation demonstrates promising results, which backed by solid theoretical analysis. The paper is well written.

However, I still have several questions and concerns as follows:

* *Overall Memory Savings* - It's a bit misleading to report memory savings just for activations (Table 3). The bottomline number should be overall memory savings including other objects such as input data. Also, in Section 5.3, the authors claim EXACT reduces the memory usage in training GraphSAGE on obgn-products to fit in 11GB. However, a sizable portion of memory savings come from AMP, which is orthogonal to EXACT. The authors should separately report memory savings of EXACT from those of AMP (and other techniques, if any).

* *Scalability* - How effective would EXACT be if node-wise or layer-wise sampling methods were used in a mini-batch setting?  Would input data may take a dominant portion of memory usage once the activation map shrinks after aggregation? Also, how scalable would EXACT be for larger (e.g., billion-scale) datasets and/or deeper GNNs?

* *Design Rationale* - EXACT applies random projection first, followed by quantization during the forward pass. What if we switch the order? Is there any convincing reason why RP-followed-by-quantization is preferred over quantization-followed-by-RP?

* *Insight into Activation Compression* - The GNNs in this work are very resilient to activation compression with quite small accuracy loss regardless of their depths. Can you provide some insight into why GNNs are so robust against this somewhat extreme compression of activations?

Nits:

* (Section 5) "it applies the random projection *following by* a 2-bit quantization" -> followed by
* (Section 5.2) "*From Table 3*, the overhead of EXACT (INT2) is roughly 12% ∼ 25%." -> "From Figure 3, ..."


**Summary Of The Paper:**

This paper proposes EXACT, a framework for training GNNs with compressed activations with two methods: quantization and random projection. The primary design objective of EXACT is to trim down the memory consumption in GNN training while maintaining acceptable training speed and accuracy. EXACT achieves significant memory savings with 0.2-0.5% accuracy drop and 10-25% slowdown of training throughput across five models and five graph datasets.

**Summary Of The Review:**

This paper introduces a simple yet effective technique to reduce the memory usage in GNN training, which demonstrates promising results. However, I still several questions/concerns for evaluation metric, scalability, design rationale, and key insight. I'd ask the authors to address them in their rebuttal.

---

> ### Author Response · Authors · 2021-11-18
> **Response to Reviewer eNSi [part 1/3 Q1]**
>
> Thanks for your insightful comments. Below, we provide detailed responses to your concerns.
>
> **Q1. [Overall Memory Savings and analysis regarding AMP]**
>
> **Overall memory saving ratio:**
> We thank the reviewer for pointing out this problem. In the updated version, we add a more detailed analysis of the memory usage of each part, including activation maps, input data, and activation gradients (Table 18 and Table 19). Generally, the overall compression ratio ranges from 1.5X to 18X.
>
> Here we want to justify that we prefer to report the memory usage of activations instead of the overall memory usage because **the overall memory saving ratio can be improved in the (near) future**. Currently EXACT is implemented for PyG. For PyG, the memory usage of input data can be greatly reduced: Until the time of our submission, PyG stores the graph structure data (mainly the node ID and edge ID) in int64/long data type, and cannot be changed.  We note that for almost all commonly-used node-classification datasets, the int32 data type is enough since it can represent an integer number up to two billion (four billion with uint32). Currently, the largest node classification dataset in OGB contains 244M nodes, 1.7B edges (OGB-MAG240M). We can safely cast them into the int32 data type without any loss. Considering that DGL has supported storing the graph in int32 (https://docs.dgl.ai/en/0.6.x/generated/dgl.DGLGraph.idtype.html#dgl.DGLGraph.idtype), we generally think PyG will support the int32 data type for the graph structure data soon. Thus, the overall memory saving ratio can be greatly improved.
>
>
> In fact, our SpMM implementation support both int32/int64 graph structure data. For the compatibility concern, we do not leverage this feature in the benchmark.
>
> **AMP**:
> For AMP, the results are as below:
>
> |                 | Data Mem (MB) | Act Mem (MB) | Peak BWD Mem (MB) |
> |:---------------:|:-------------:|:------------:|:-----------------:|
> |     Baseline    |      4811     |     16555    |        6176       |
> |       AMP       |      4811     |     9379     |        3500       |
> | EXACT w./o. AMP |      4811     |      572     |        6176       |
> |  EXACT with AMP |      4811     |      572     |        3500       |
>
>
>
> “Peak BWD Mem” means the peak memory usage during the backward pass, which encompasses the activation gradients and other work space memory.
>
> We want to clarify that the main reason we employ AMP is to compress the Peak BWD Mem (see Appendix H “Analyzing the memory usage”). **For EXACT, the compression ratio of activation maps will not be affected by AMP.**  This is because each float16/float32 number will finally be compressed into a 2-bit integer, regardless of the bit-width before compression. However, with AMP, the memory usage of activation gradients will be compressed to roughly half the original size since we store the gradients in half precision (6176->3500). Without AMP, the data mem + peak BWD mem =  10978MB, which already approaches the physical limit of 11GB memory. We need AMP to leave more space for running the algorithm: With AMP, data mem + peak BWD mem=8311, there is roughly 2953 MB left. We use EXACT to pack the 9379MB (w./ AMP) activation map into the left space (we do not consider the work space memory here).

---

> > ### Comment · Reviewer_eNSi · 2021-11-30
> > **Response to authors**
> >
> > Thank you for your careful response to my questions and concerns. I have read it carefully and think that my questions are adequately addressed in the rebuttal and revision. However, I eventually have decided to stick with my original score after the rebuttal. While I still appreciate this work tackling an important problem and demonstrating promising results in saving memory footprint for GNN training, the novelty concern raised by Reviewer 2GtZ made me less convinced of the main intellectual contributions of this work.

---

> ### Author Response · Authors · 2021-11-18
> **(Continued) Response to Reviewer eNSi [part 2/3 Q2]**
>
> **Q2. [the memory usage of input data in other samplers and scalability]**
>
> > Would input data may take a dominant portion of memory usage once the activation map shrinks after aggregation?
>
> We think this case will rarely happen. The high-level idea is that the structure of the sampled subgraph will not be far from the whole graph. Also, the memory ratio of input data to activation maps is decided by the subgraph structure/sparsity. Thus, the input data will not take a dominant portion of memory usage. We performed a case study for the widely used node-wise sampler, SAGE sampler.  The model is a 3-layer 256-wide GraphSAGE, and the dataset is ogbn-products. For the hyperparameter, we follow the official tutorial in PyG (https://github.com/pyg-team/pytorch_geometric/blob/master/examples/ogbn_products_sage.py). The batch size is 1024 and the number of neighbors to sample is set to 15, 10, 5 for the three layers, respectively. The results are as follows:
>
> Data Mem: 147.7, Act Mem: 414, Peak BWD Mem: 156.
>
> Also, as we mentioned in the response to Q1, the memory usage of input data can be greatly reduced.
>
>
> **Scalability**:
> There are mainly three parts occupying the memory: the input data, activation maps, and the activation gradients. EXACT can only compress the activation maps. In the single GPU setting, as we analyzed in the second part of Q1, if the sum of the other two parts approaches the physical limit of a single GPU, then EXACT cannot train the model. This is also the main limitation of EXACT. However, we want to point out that for any activation compression algorithm, this statement is also true.
> For the billion-wise graph dataset, we do not think we can train the model with EXACT using a single GPU. For example, for the ogbn-papers100M, the input data consumes roughly 51GB (with PyG), which already surpasses the physical limit of most of the GPUs. In this case, we can train the model only with multi-GPUs. As we pointed out in the main paper, EXACT can be extended to the multi-GPU setting. And the main value of EXACT in the multi-GPU setting is to trim down the number of GPUs you need. We leave it as future work.
>
> For deeper GNN models, as shown in Table 18 and Table 19, we think EXACT is particularly useful in this case. This is because when we increase the model depth, the ratio of activation maps will significantly increase (see GCNII in Table18 and its surrounding explanation). Here we provide a case study for scaling up the depth of GCNII on the ogbn-arxiv dataset using a single GTX 1080 (11GB) GPU. We do not change any hyperparameters except the model depth. According to Table 12, we set $\frac{D}{R}=8$ for EXACT (RP+INT2).
> Below we report the maximum depth which does not raise an OOM error. We summarize that in this case study, EXACT can scale up the depth up to $30.3\times$ without changing the hardware.
>
> Baseline: 9 layers
>
> EXACT (INT2): 142 layers (15.7X)
>
> EXACT (RP+INT2): 273 layers (30.3X)

---

> ### Author Response · Authors · 2021-11-18
> **(Continued) Response to Reviewer eNSi [part 3/3 Q3 and Q4]**
>
> **Q3. [Design Rationale: why RP-followed-by-quantization]**
>
> In practice, we cannot exchange the order of these two operations due to the practical platform limitation. First, we note that Pytorch currently does not support data types below one-byte due to some engineering trade-off (see https://github.com/pytorch/pytorch/issues/41571 for an explanation). Hence, to maximize the memory saving, we actually hack the data format in Pytorch. For example, if we want to store the tensor using int2 format, we actually pack four int2 numbers into one int8 number, where int8 is a built-in data type in Pytorch. Thus, after quantization, the tensor is actually a highly compressed 8-bit bit-stream. Since memory is byte-addressable, without decoding the bit-stream, each 8-bit number in this bit-stream does not make any sense. Thus in practice, we cannot apply the random projection after the quantization.
>
> **Q4. Insights into Activation Compression**
>
>
> We respectfully correct the misunderstanding that the GNNs in this work are very resilient to activation compression with quite a small accuracy loss regardless of their depths. We point out here that the deepest GNN in our paper is 16-layer, which is still considerably shallower than CNNs.
> We show in Appendix E.4 that the approximation error will compound layer-by-layer during the backward pass. Here we think the key insight behind the theory and observation is that "since the variances introduced by compressed contexts at different layers will accumulate,  it suggests that the noise introduced by the compressed context is relatively small for shallow models" (quote from Appendix E.4).
> We experimentally investigate how many layers can be regarded as "shallow" in the sense that the approximation error will not lead to a significant accuracy drop.
> Our GCNII experimental results suggest that the approximation error is still not large enough for deeper GNNs (up to 16-layer).
> The reasons we do not further go deeper are two folds **(1)** it is still one open challenge to build deeper GNNs. According to [1], for ogbn-arxiv (the largest dataset in [1]), most deep GNNs perform the best at 16-layer. When the depth goes up to 32-layer, there is a performance drop. **(2)** Deeper GNNs run significantly slower, which is empirically verified in Figure 3. Up to our knowledge, currently, the deepest GNN can have 1000 layers [2]. It requires 17.1 days to train the model using a single V100 GPU [2], which is unaffordable for us. We leave examining EXACT with deeper GNNs as future work.
>
> [1] Bag of Tricks for Training Deeper Graph Neural Networks: A Comprehensive Benchmark Study
>
> [2] Training Graph Neural Networks with 1000 Layers

---

### Official Review · Reviewer_2GtZ · 2021-11-02

**Correctness:** 3
**Technical Novelty And Significance:** 2
**Empirical Novelty And Significance:** 2
**Recommendation:** 3
**Confidence:** 5

**Main Review:**

**Strengths**:
- I think the work is solving an important problem -- although I would note that we have existing works in this area, which solve the problem pretty well.
- The results are good, especially when considering quantization only.
- The authors did a good job choosing datasets -- they go up to OGB-products, which is nice.
- The choice of models is good -- and I appreciate that the authors included GCNII to evaluate how their method scales with depth. I'd be delighted if they added GAT as well, though :-) -- that's really part of the reason we want their method!

**Weaknesses:**
- I am not entirely sure what the benefit of this approach is compared to sampling or historical embedding approaches is. Could you explain what is the benefit of this approach over others?
- It's not really clear what the benefit of random projections are, given that they cause significant accuracy degradation. I am aware that they do improve the compression ratio, but I would argue that in most cases there's little incentive to use it.
-If we consider just quantization, then there is little novelty to this work I would argue: I am not sure what is novel over the Chakrabati paper at NeurIPS 2019 is. I am aware that they focused on CNNs, but there is little reason to believe that their method would not work for GNNs.

In fact, I did actually implement the Chakrabati method myself for GNNs -- and unsurprisingly it does work. I never wrote up the results (!) or studied it too carefully. It's useful that this paper has done that though. I am unsurprised it is effective down to 1-bit integer by the way -- the gradients are much "smoother" for GNNs than for CNNs, as the effective batch size is very large.

- There is an interesting work that was accepted to NeurIPS this year that you should compare to: "AC-GC: Lossy Activation Compression with Guaranteed Convergence"; they look at CNNs and Transformers, but there's no reason why you can't apply their technique to graphs as far as I can tell.

It is unfortunate that my review seems rather negative. I actually don't have a particularly negative view of the paper: I just have many questions (and unfortunately, they are related to novelty). I don't think that they're unanswerable, and I consider myself to be pretty reasonable at rebuttal if you can present new evidence / answer some questions.

I would make the case that novelty is not the absolute most important thing -- but I am a little skeptical of the contributions to the literature presented by this work. However, the work is likely to be of good use to practitioners -- it's not difficult to implement many parts of this, and the results do seem impressive.

**Summary Of The Paper:**

This work attempts to train GNN models with reduced memory requirements. Normally when training, we need to store activations for calculating the gradients on the backwards pass, and we typically keep these at full (32-bit) precision. This work argues that we don't need to do this, and instead we can significantly reduce the memory footprint at training time by keeping low precision activations for the backwards pass. They use two approaches for this: quantization, and random projections

**Summary Of The Review:**

I am leaning towards a reject for now, but I am happy to be convinced. I am aware of how crushing it can be to authors to read these things -- and I would like to say that I think the work has strong value to the community -- but it is not necessarily a strong piece of novel research.

Post rebuttal:

I am not convinced that my original view has changed regarding novelty. I strongly recommend that the AC rejects the paper since I do not believe the research contribution is sufficient.

---

> ### Author Response · Authors · 2021-11-18
> **Response to Reviewer 2GtZ [part 1/3 Q1 and Q2]**
>
> **Q1. [Benefits over sampling or historical embedding approaches]**
>
> As we mentioned in Appendix B “related work and discussion​”, we would like to view our approach as an orthogonal technique to sampling or historical embedding.
> Considering that the real-world graphs can contain billions of nodes (e.g., social networks and recommender systems), it is always impossible to leverage a standalone approach to train GNNs at such a scale.
> As we pointed out in Appendix B, EXACT can be applied over these two approaches:
> for subgraph sampling, EXACT can enlarge the maximal batch size. For GNNAutoScale (historical embedding), EXACT may reduce its time overhead since the quantization is considerable faster than swapping data between CPU and GPU.  We experimentally demonstrate that our approach could be applied over the sampling methods to further improve the memory efficiency (Table 3) or improve accuracy by scaling up the batch size (Appendix I.5).
>
> **Q2. [Benefits of random projection]**
>
> We would like to point out that the benefits of random projection are two folds. **(1)** From the application perspective, we agree that for commonly used shallow models, the benefit from random projection is limited, especially when we consider the overall memory saving ratio. However, as we show in Table 16 and Table 17, for deep GNN models, the memory saving from random projection is non-trivial. For example, for GCNII on ogbn-arxiv, with the help of random projection, we further improve the maximal depth from 142 layers (int2 quantization only) to 273 layers (Random Projection + int2 quantization). The detailed experimental setting is listed in our response to Reviewer eNSi Q2. Also, we note that in the OGB large graph challenge, one winning solution utilizes deep GNN up to 50-layer [1]. We think EXACT can help trim down its hardware requirements. **(2)** From the research perspective, as we mentioned in the paper, if GNN can tolerate the extreme low-bit quantization, then one valuable open question is that how can we go further beyond one-bit? We make an initial attempt to answer the question by combining the random projection and vanilla quantization. As we mentioned in the introduction, random projection and quantization have an “interaction” effect: First, incorporating random projection does not introduce or even reduce the overall time overhead.
> This is because, after random projection, the total number of elements to be quantized is reduced by a factor of $\frac{D}{R}$.
> Second, as suggested by Proposition 3 and experimental results, although the accuracy drop of random projection is larger than quantization, a 0.5% performance drop after incorporating random projection is still acceptable.

---

> ### Author Response · Authors · 2021-11-18
> **(Continued) Response to Reviewer 2GtZ [part 2/3 Q3]**
>
> **Q3. [limited novelty if we ignore the random projection & limited contribution]**
>
> For the value of random projection, please see our response to Q2.
>
> **Empirical contributions to the community:**
> Before discussing the novelty of EXACT, we would like to reiterate the importance of the time overhead and memory saving ratio.
> Namely, they are often the barrier to the practicality of compression methods.
> With this in mind, we think the engineering efforts are underestimated due to the following reasons.
> Up to our knowledge, directly extending BLPA (i.e, the Chakrabati paper at NeurIPS 2019) [2] to GNNs will raise sub-optimal memory saving ratios and hardware throughput.
> This is caused by lacking the optimization to the ReLU, Dropout, and many micro-level details.
> We apologize that we have not released our code in the initial submission.
> Our code is available at https://anonymous.4open.science/r/exact-revd.
> We note that our code is ready to use for many common graph related tasks.
> Below we briefly list the common and different parts of previous work, which also serve as the acknowledgment and the guideline to our low-level CUDA kernels. A more detailed version can be found in the README file.
>
>
> 1. For quantization/dequantization,  BLPA [2] and ActNN [3] hack the byte format in the same way to maximize the memory saving. For example, if the precision is int2, we pack four int2 numbers into one int8 number.
> To reduce time overhead, we simplify the per-group quantization kernel of ActNN to the vanilla quantization.
>
> 2. To balance the trade-off between time overhead and memory saving ratio, we fuse the quantization kernel with the ReLU and Dropout kernel.
> Here we utilize ActNN's ReLU implementation since its speed is near-optimal compared to the official one. We extend the implementation to LeakyReLU and ELU for GAT.
> We note that ActNN also has one quantized dropout function. However, it runs too slow such that the overall time overhead will even enlarge up to 40% with their implementation.
> Here we provide our dropout CUDA kernel, whose speed is near-optimal compared to the official one.
>
> 3. We extend the PyG SpMM kernels such that to support both int32/int64 data types. That said, you can store the graph data using int32 data type (if both #nodes and #edges <= 2^31 - 1) and train the model with our SpMM kernel. We highly recommend enabling this feature when you have less than 11GB of GPU memory.
>
>
> **Novelty/New Insights:**
> We would like to point out that we are not merely reporting the observation that the 1-bit/2-bit quantization works for GNNs.
> Here we discuss the insights and implications behind this observation in more detail.
> From the numerical analysis [6], the jacobian matrix (gradient) between two tensors can be viewed as a sum over all "paths" connecting these two tensors (see Figure 1 in [6], where we essentially change the scaler to tensor for neural networks).
> Moreover, the architecture of GNNs is considerably shallower than CNNs and transformers.
> Thus, for GNNs, the key intuition is that these paths are significantly "shorter" compared to those of CNNs. As a result, the approximation error (i.e., the variance from quantization) along the path is hard to get accumulated (Appendix E.4). Hence, EXACT works well for GNNs even at an extreme compression ratio.
> Inspired by the above insight, we also experimentally investigate how many layers can be regarded as "shallow" in the sense that the approximation error will not lead to a significant accuracy drop through the ablation study of GCNII. Our GCNII experimental results suggest that the approximation error is still not large enough for deeper GNNs (up to 16-layer).
> We think this explanation is more accurate than the comment "the gradients are much smoother for GNNs than for CNNs, as the effective batch size is very large" in the sense that the int2 quantization also works well for subgraph sampling (e.g., for Yelp, as implied by Table 12, the number of nodes in the sampled subgraph of GraphSAINT is roughly 2 x 1250 = 2500, which is considerably smaller than the full batch one).
>
> From the technique perspective, we agree that our quantization method follows the framework proposed in BLPA.
> However, this is not our novelty/contribution.
> In fact, most of the activation compression methods follow the framework proposed in BLPA, and the main challenge is how to select the best bit-width for each layer/sample.
> The most valuable part of our work (and thus the novelty) is that, we mathmatically and experimentally demonstrate that the simplest 1/2 bit quantization is enough for GNNs. Other complex methods, such as mixed precision quantization, are overkill.
> More valuable research questions are how can we further go below one-bit for GNNs.
> As we mentioned in our response to Q2, we make an initial attempt to this question by combining random projection and quantization.

---

> ### Author Response · Authors · 2021-11-18
> **(Continued) Response to Reviewer 2GtZ [part 3/3 Q4]**
>
>
> **Q4. [Comparison against AC-GC]**
>
> Thank you for pointing it out. We add AC-GC as one of the references. Since the Neurips 2021 camera-ready paper is online on Nov 9 (we did not find a pre-print version of AC-GC), so we have not tried the official code of AC-GC. We respectfully refuse to add the AC-GC comparison experiment for the following reason. From the paper, we think AC-GC tries to automate the bit-width selection process, which is important to minimize the accuracy drop for CNNs and transformers. We agree that AC-GC can be extended to GNNs and can achieve a near-lossless result. However, as we mentioned in the response to Q3, the performance drop of vanilla 1-bit/2-bit quantization is acceptable/negligible. Thus, when we consider the time overhead, we think AC-GC is technical overkill for GNNs. Up to our knowledge, AC-GC also cannot surpass the physical one-bit limitation. From our mathematical and experimental analysis, for GNNs, instead of searching bit-width and mixed-precision quantization, a more valuable research direction is how to compress activations below one-bit and/or jointly compress the graph structure data.
>
> Reference
>
> [1] Large-scale graph representation learning with very deep GNNs and self-supervision
>
> [2] Backprop with Approximate Activations for Memory-efficient Network Training
>
> [3]  ActNN: Reducing Training Memory Footprint via 2-Bit Activation Compressed Training
>
> [4] Gist: Efficient Data Encoding for Deep Neural Network Training
>
> [5] MONeT: Memory Optimization for Deep Networks
>
> [6] Computational graphs and rounding error

---

> > ### Comment · Reviewer_2GtZ · 2021-11-21
> > **Response to authors**
> >
> > > we would like to view our approach as an orthogonal technique to sampling or historical embedding
> >
> > This is a valid point. My argument was more to do with how effective the method is relative to these in a standalone sense. You have not really answered this to my knowledge.
> >
> > > This is consistent with previous finding that the larger batch size does not always lead to better performance
> >
> > This is a quote from I.5, serving as a counterpoint to the points you are making in this rebuttal.
> >
> > > a 0.5% performance drop after incorporating random projection is still acceptable.
> >
> > I think this is a stretch -- 0.5% is quite a lot to lose. How are we defining "acceptable"?
> >
> > Moving onto Q3, one major argument is that you've written CUDA kernels which implement the idea in a way that the whole community can use. I am not sure how much of a "research" contribution this really is (although I accept that this is a useful contribution in a practical sense). I leave it to the AC to decide how much value they place on this.
> >
> > > We think this explanation is more accurate than the comment "the gradients are much smoother for GNNs than for CNNs, as the effective batch size is very large" in the sense that the int2 quantization also works well for subgraph sampling
> >
> > It can be both factors. It would be unwise to discount the argument I have provided without any evidence.
> >
> > > The most valuable part of our work (and thus the novelty) is that, we mathmatically and experimentally demonstrate that the simplest 1/2 bit quantization is enough for GNNs.
> >
> > This seems tenuous to me. The fundamental issue I have is that this does not seem like a "surprising" result to me. I still don't believe that there is any reason for us to be surprised that methods like BLPA work acceptably for GNNs.
> >
> > It's worth stating again: "value" is not the same as "novelty". It is up to the AC to decide how they want to balance this.
> >
> > > We respectfully refuse to add the AC-GC comparison experiment for the following reason
> >
> > OK, I can accept the reasoning for not doing this right now. However, I would kindly ask the authors if their paper is accepted that they add the comparison for camera ready if the code is released.
> >
> > > Thus, when we consider the time overhead, we think AC-GC is technical overkill for GNNs
> >
> > My understanding is that the time overhead is not too bad. Also, despite it being "overkill", the main benefit of their method is that you don't really need to tune much, and it will automatically discover the correct parameters.
> >
> > > Up to our knowledge, AC-GC also cannot surpass the physical one-bit limitation
> >
> > This is not true -- the method is agnostic to the compression backend. Hence you can use approaches such as JPEG or CuSZ to do the compression.
> >
> > To be clear: I have no affiliation with this paper -- I just think it's a good paper, and it's important for this work to compare to it adequately.
> >
> >
> >
> > I am going to stay at reject, since my questions were not answered adequately. The major contribution of this work is showing that methods that work for other NN architectures work also work for GNNs is not -- in my opinion -- enough to publish at a competitive venue like ICLR. It is up to the AC to decide if they have a different perspective, but I am not convinced by the novelty of this submission.

---

> > > ### Author Response · Authors · 2021-11-23
> > > **Further Responses to Reviewer 2GtZ [part1/3]**
> > >
> > > We really appreciate your time and constructive suggestions.
> > > We apologize that we did not directly answer the questions you raised in the first round rebuttal.
> > > Below we will try our best to answer the questions.
> > >
> > >
> > > > This is a valid point. My argument was more to do with how effective the method is relative to these in a standalone sense. You have not really answered this to my knowledge.
> > >
> > > We apologize that we misunderstood your point in Q1 and here we have conduct the comparision between EXACT and subgraph sampling methods in a standalone way.
> > > For GNNAutoScale [1], it also changes the low-level implementation of PyG [1].
> > > Due to time restriction, we still haven't completely gone through their codebase and integrated theirs with our codebase. For this request, we will add the comparison in the final version.
> > >
> > > We compare full batch GraphSAGE with EXACT to subgraph sampling methods under a fixed activation memory budget (**Appendix I.5.1**). The results are shown in the following tables. The detailed experimental settings can be found in **Appendix I.5.1**.
> > > For Reddit and Flickr, full batch training with EXACT can outperform subgraph sampling methods.
> > > For Yelp, subgraph sampling methods are better than full batch training with EXACT.
> > > However, instead of EXACT,  we note that this gap comes from the full batch training setting.
> > > As shown in Table 3, without EXACT, the F1-micro of full batch GraphSAGE is $62.05\%$, which is much lower than that of subgraph sampling methods.
> > > In summary, subgraph sampling methods may outperform the full batch training on some datasets (e.g., Yelp).
> > > For these datasets, we think full batch training with EXACT cannot outperform subgraph sampling methods.
> > > For other datasets, as shown in the following tables, full batch training with EXACT can outperform subgraph sampling methods under a fixed activation memory usage.
> > >
> > > | Dataset | Method                               | Accuracy (%) / F1-micro |
> > > | :------ | :----------------------------------: | :---------------------: |
> > > | Flickr  | GraphSAINT                           | 50\.30 ± 0.16           |
> > > |         | Cluster-GCN                          | 49\.98 ± 0.15           |
> > > |         | Full Batch  <br> w./ EXACT (INT2)    | 51\.97 ± 0.20           |
> > > |         | Full Batch <br> w./ EXACT (RP+INT2)  | 51\.83 ± 0.21           |
> > >
> > > | Dataset | Method                               | Accuracy (%) / F1-micro |
> > > | :------ | :----------------------------------: | :---------------------: |
> > > | Reddit  | GraphSAINT                           | 96\.18 ± 0.03           |
> > > |         | Cluster-GCN                          | 96\.03 ± 0.05           |
> > > |         | Full Batch <br> w./ EXACT (INT2)     | 96\.40 ± 0.05           |
> > > |         | Full Batch <br> w./ EXACT (RP+INT2)  | 96\.34 ± 0.03           |
> > >
> > > | Dataset | Method                               | Accuracy (%) / F1-micro |
> > > | :------ | :----------------------------------: | :---------------------: |
> > > |  Yelp   | GraphSAINT                           | 63\.20 ± 0.19           |
> > > |         | Cluster-GCN                          | 63\.26 ± 0.14           |
> > > |         | Full Batch <br> w./ EXACT (INT2)     | 61\.95 ± 0.12           |
> > > |         | Full Batch <br> w./ EXACT (RP+INT2)  | 61\.59 ± 0.12           |
> > >
> > > > (Author) This is consistent with previous finding that the larger batch size does not always lead to better performance.
> > > (Reviewer 2GtZ) This is a quote from I.5, serving as a counterpoint to the points you are making in this rebuttal.
> > >
> > > Thank you for pointing it out.
> > > One common belief in the graph community is that the full-batch training is better than subgraph sampling methods since the subgraph may fail to preserve the edges that present a meaningful topological structure [1,5,6].
> > > However, as examined in our paper and OGB [5], this common belief is not true for ogbn-products.
> > > Moreover, from our experiments, the $3\times$ large batch size performs the best on ogbn-products.
> > > Thus, for some datasets,
> > > there may exist an optimal batch size for subgraph sampling methods.
> > > Since the memory of the hardware may not be enough to train models with such an optimal batch size, by reducing the memory usage of activations,
> > >  EXACT enlarges the "search space" and thus allows the optimal batch size to be discovered. We update **Appendix I.5.2** to reflect this message.

---

> > > ### Author Response · Authors · 2021-11-23
> > > **(Continued) Further Responses to Reviewer 2GtZ [part2/3]**
> > >
> > > > (Author) a 0.5% performance drop after incorporating random projection is still acceptable.
> > > (Reviewer 2Gtz) I think this is a stretch -- 0.5% is quite a lot to lose. How are we defining "acceptable"?
> > >
> > > We apologize for using  the hand-waving term, "acceptable", without directly citing its source.
> > > We would like to cite that in other works, a $0.5\%$ accuracy drop is often regarded as "negligible". Below we quote some sentences from references to support our claim.
> > >
> > > 1. (ActNN) "ActNN compresses activations to 2 bits, with negligible (< 0.5%) accuracy loss." [2]
> > >
> > > 2. (BLPA) "Our algorithm enables up to 8x memory savings, with negligible drop in training accuracy compared to exact training". From Table 1 in [3], we summarize that their 4-bit quantization results are CIFAR-10 $\downarrow0.13\%$, CIFAR-100 $\downarrow0.14\%$, and ImageNet $\downarrow0.52\%$ [3]
> > >
> > > 3. (OGB) "Table 4 also shows that the recent mini-batch-based GNNs give promising results, even slightly
> > > outperforming the full-batch version of GraphSAGE that does not fit into ordinary GPU memory" According to Table 4 in [5], Cluster-GCN ($78.97\%$) GraphSAINT ($79.08\%$) are "slightly outperforming" full-batch GraphSAGE ($78.50\%$). Here the accuracy gaps are $0.47\%$ and $0.58\%$ for Cluster-GCN and GraphSAINT, respectively.
> > >
> > > It is under this context we described $\approx 0.5\%$ accuracy drop to be "acceptable".
> > >
> > > > It can be both factors. It would be unwise to discount the argument I have provided without any evidence.
> > >
> > > We apologize for the lack of rigorous evaluation in the sense that we simply use the Yelp example to support our claim.
> > > We think our theoretical results in Appendix E.4 can be made more solid by investigating the role of batch size to activation compressed training.
> > > We scale down the batch size of both GraphSAINT and Cluster-GCN to only 500. Then we apply INT2 quantization over these two subgraph sampling methods.
> > > We add the results to the updated version (**Table 6 and Table 7 in Appendix E.4**).
> > > Here we summary that INT2 quantization can even achieve near-lossless results under a much smaller batch size.
> > > A more detailed analysis can be found in **Appendix E.4**.
> > > From these additional results, we think the main reason behind the phenomenon is GNN's much shallower architecture.
> > >
> > > > (Author) Up to our knowledge, AC-GC also cannot surpass the physical one-bit limitation.
> > > (Reviewer 2GtZ) This is not true -- the method is agnostic to the compression backend. Hence you can use approaches such as JPEG or CuSZ to do the compression.
> > >
> > > We apologize for our misunderstanding in the first round rebuttal.
> > > Given the context of comparing EXACT to AC-GC, we thought both EXACT and AC-GC should utilize the quantization to compress the activation.
> > > Thus, what we mean here is "**for quantizing activations**, AC-GC also cannot surpass the physical one-bit limitation".
> > > After carefully going through the AC-GC paper,
> > > we agree that AC-GC can go below one-bit with CuSZ and JPEG-ACT [4].
> > > And it is meaningful to examine CuSZ, JPEG-ACT, and AC-GC for GNNs.
> > > Due to the time limit, we apologize that we do not examine them in the current version.
> > > We discuss the difference between EXACT and AC-GC in **Appendix B**.
> > > **Generally, we think EXACT and AC-GC approach the problem from different perspectives.**
> > > For EXACT, we value the compression rate more than the accuracy change.  Namely, given a preset memory budget, we try to fit the model with acceptable ($\approx 0.5\%$) accuracy drop.  For AC-GC, it values the accuracy change more than the compression rate in the sense that the compression rate is adapted to the given allowable accuracy drop.
> > > We think both of them have their own important applications and we will compare them accordingly in the final version.

---

> > > ### Author Response · Authors · 2021-11-23
> > > **(Continued) Further Responses to Reviewer 2Gtz [part3/3] (Discussion about Novelty)**
> > >
> > > > Moving onto Q3, one major argument is that you've written CUDA kernels which implement the idea in a way that the whole community can use. I am not sure how much of a "research" contribution this really is (although I accept that this is a useful contribution in a practical sense). I leave it to the AC to decide how much value they place on this.
> > >
> > > We would like to clarify that we have explicitly stated that this part is not to justify the novelty of our work. As we mentioned in the first sentence in the subsection "Empirical contributions to the community" in Q3, "**Before discussing the novelty of EXACT, ...**",
> > > the purpose of this subsection is to show "**we think the engineering efforts are underestimated**" to address the comment " it's not difficult to implement many parts of this."
> > >
> > > **Discussion about our work's novelty:**
> > >
> > > According to Q2 and Q3,
> > > the random projection part in EXACT is excluded when discussing the novelty.
> > > And the reason is that the random projection causes "significant" accuracy degradation ($\approx 0.5\%$).
> > > However, we remark it is incorrect to ignore the random projection when discussing our work's novelty due to the accuracy drop.
> > > According to previous works, a $\approx 0.5\%$ accurancy drop (or increase) can be regarded as "negligible" [2,3] (or "slightly outperforming" [5]). Moreover, it is worthwhile to trade such an accuracy degradation for reduced memory usage since,  given a fixed memory budget, EXACT(RP+INT2) enables the usage of deeper GNN models on larger datasets.
> > > Namely, although there may be a larger accuracy drop (usually $\leq0.5\%$) after incorparating random projection, the accuracy drop can be compensated by the stronger expressiveness of deeper GNNs.
> > > As we can see from Table 3,
> > > **GCNII with EXACT (RP+INT2) still outperforms all other shallow baselines across different datasets.**
> > >
> > > We apologize for not stating our novelty in a clearer manner in the first round rebuttal.
> > > Since we think the importance of random projection is mentioned in our response to Q2, we did not include it when discussing the novelty in our response to Q3.
> > > However, we remark that the value of random projection is indispensable when discussing the novelty of our work.
> > > Below we collect the discussions regarding our work's novelty in one place.
> > >
> > >
> > > * (Novelty 1) We theoretically show GNN is more noise-tolerant to compressed activations than CNN due to its shallow structure.
> > > Moreover, our theoretical analysis is **agnostic** to compression methods.
> > > To the best of our knowledge, no existing literature shows this connection and systematically investigates (1) why GNN is more noise-tolerant to compressed activations; (2) how many layers can be regarded as "shallow" in the sense that the approximation error will not lead to a significant accuracy drop.
> > >
> > > * (Why random projection is important)
> > > We remark that this bullet item is not to justify our novelty.
> > > However, we include it to provide the context for a fair judgment when discussing the novelty of our work.
> > > As we mentioned in our response to Q2 (i.e., the quantitive results about the overall memory saving ratio), random projection is crucial for building deeper GNNs.
> > > We remark that the random projection has an important use case of enabling deeper GNNs within a fixed memory budget (e.g., 5GB or 12GB GPU memory).
> > > Although there may be a larger accuracy drop (usually $\leq0.5\%$) after incorparating random projection, the accuracy drop can be compensated by the stronger expressiveness of deeper GNNs.
> > > As we can see from Table 3, **GCNII with EXACT (RP+INT2) still outperforms all other shallow baselines across different datasets.**
> > >
> > > * (Novelty 2) We theoretically show random projection and quantization can have an **interaction** effect (the whole is greater than the sum of its parts).
> > > Namely, from the model performance aspect, after random projection, applying
> > > quantization only has a limited impact on the model performance (Proposition 3). From the time aspect, EXACT
> > > runs comparable or even faster than quantization only (Figure 3).
> > > This is because, after random projection, the total number of elements to be quantized is reduced by a factor of $\frac{D}{R}$.
> > > To the best of our knowledge, no existing work shows such theoretical results and
> > > applies such extreme compression strategy for compressing activations.
> > > We also kindly point out that all theoretical results are directly supported by corresponding experiments (Proposition 1-3).
> > >
> > > [1] GNNAutoScale: Scalable and Expressive Graph Neural Networks via Historical Embeddings
> > >
> > > [2]  ActNN: Reducing Training Memory Footprint via 2-Bit Activation Compressed Training
> > >
> > > [3] Backprop with Approximate Activations for Memory-efficient Network Training
> > >
> > > [4] JPEG-ACT: Accelerating Deep Learning via Transform-based Lossy Compression
> > >
> > > [5] Open Graph Benchmark: Datasets for Machine Learning on Graphs
> > >
> > > [6] Improving the Accuracy, Scalability, and Performance of Graph Neural Networks with Roc

---

> > > ### Author Response · Authors · 2021-11-29
> > > **Looking forward to your response and discussion**
> > >
> > > Dear Reviewer 2Gtz,
> > >
> > > We appreciate your time and constructive suggestions. We apologize that we did not directly answer some of your questions in our first round rebuttal.
> > >
> > > We sincerely hope to have further discussion with you to see if our further response solves your concerns, since **today** is the deadline of final stage of discussion period.
> > >
> > > Although we cannot update our draft due to ICLR policy, we are happy to answer any additional questions and provide more information.
> > >
> > > Thank you.
> > >
> > > Best wishes,
> > >
> > > Authors

---

> > > > ### Comment · Reviewer_2GtZ · 2021-11-29
> > > > **Response to authors**
> > > >
> > > > I am still not keen to raise my score. The novelty remains an issue, and you remain unable to provide a convincing response in this direction. It remains a big issue for me, and this work is not adding much to our knowledge as a community in my opinion. I am raising my confidence to 5 now as well -- I would strongly argue to the AC to consider the value of this contribution relative to the existing literature.
> > > >
> > > > The primary result of this paper -- that GNNs are tolerant to compressed activations -- is not a surprising result. Although I appreciate the theoretical analysis, I am not sure I consider it to be enough to constitute enough novelty for the work to be accepted. The second claimed result, regarding the interaction effect, is interesting, but you are overstating the benefit of random projection relative to existing works.

---

> ### Author Response · Authors · 2021-12-06
> **Outline of our rebuttal**
>
> First of all, we apologize for the delayed reply. Although we have different opinions on the novelty of this work, I am glad that we have reached a consensus on the value/practical contribution of this work:
>
> 1. This work has a strong practical contribution to the community (i.e., the optimized CUDA kernels)
>
> 2. The experimental evaluation is comprehensive and the experimental results are impressive
>
> 3. This work tries to solve an important problem with a simple yet effective technique
>
> However, there is also disagreement regarding the novelty of our work. Here we would like to reiterate the two research contributions of our work:
>
> 1. **[Novelty 1]** The theoretical analysis about **why** GNNs are tolerant to extremely compressed activations.
>
> 2. **[Novelty 2]** The theoretical analysis about why the random projection and quantization have an interaction effect.
>
> During our conversation, we demonstrated the implementation of EXACT is non-trivial through the released code. To the best of our knowledge, there is also no previous work which extends the activation compression technique to train GNNs on large graphs. Thus, we will focus our discussion on the novelty of our research below. Additionally, we will adress the concern about our work's contribution to the knowledge of the community.

---

> > ### Author Response · Authors · 2021-12-06
> > **Our dispute to the concern regarding the novelty 1 [par1/2]**
> >
> > Reviewer 2GtZ claims that the observation reported in our work, namely, GNNs are tolerant to extremely compressed activations, is not "surprising".
> > Reviewer 2GtZ further suggests that the observation is not surprising because "the gradients are much smoother for GNNs than for CNNs, as the effective batch size is very large" in the initial review.
> >
> > **Our rebuttal**
> >
> > 1. To address the reviewer's concern, we first use the experiment result in Table 3 and Table 14 (GraphSAINT trained on Yelp) to show that the cause behind the observation **is not the large batch size, but GNN's shallower architecture** ("for Yelp, as implied by Table 14, the number of nodes in the sampled subgraph of GraphSAINT is roughly 2 x 1250 = 2500, which is considerably smaller than the full batch one").
> >
> > 2. However, reviewer 2GtZ thinks the evidence is not enough ("It would be unwise to discount the argument I have provided without any evidence").
> > To further support our claim, we experiment on two SOTA subgraph based methods (Cluster-GCN, GraphSAINT) on three datasets (Reddit, Flickr, Yelp) with a much smaller batch size ($\approx500$ nodes per batch)
> > in the updated draft (shown in Appendix E.4). Since the accuracy drop remains below $0.2\%$ with the INT2 quantization even under a much smaller batch size, this suggests that the impact of batch size is negligible to the accuracy drop of compressing GNN's activations.
> >
> > 3. However, reviewer 2GtZ thinks our theoretical analysis is not novel enough. In this response, we want to dive deeper into the novelty of our theoretical analysis.
> > We also provide an ablation study to further show the application of our theoretical analysis.
> > ***
> >
> > **Novelty of our theoretical analysis**
> >
> > Our first research contribution is the theoretical analysis that shows GNN is more noise-tolerant to compressed activations than CNN due to its shallow architecture.
> > We remark that the theoretical analysis is **agnostic** to the compression method and it holds for GNNs as long as the following two conditions are satisfied: (1) the compression method is unbiased; (2) the depth of GNN is "moderate" (we exam up to 16-layer).
> >
> > **To the best of our knowledge, this property has not been shown in previous work. Therefore, we assume what the reviewer means by "not novel enough" is that this theoretical analysis is trivial.**
> > However, we strongly disagree that our theoretical analysis is trivial in the sense that it explains **why moderately deep GNNs are tolerant to extremely compressed activations in a principal way**.
> > As long as these two conditions are satisfied, GNN's accuracy drop resulting from the approximation error is limited, regardless of the training method (full graph/subgraph training), the model architecture (attention based/spectral convolution based GNN), the batch size (large/small batch size), and the compression method (activation compression/graph structure compression).

---

> > ### Author Response · Authors · 2021-12-06
> > **Our dispute to the concern regarding the novelty 1 [par2/2]**
> >
> >
> > **Application of our theoretical analysis**
> >
> > There are many compression methods unique to the input data of the graph dataset (the input data can be regarded as the activations to the first GNN layer, see Appendix H.3).  For example, sampling-based graph sparsification is a well-known unbiased graph compression technique [1], which deletes unimportant edges in the graph.
> > Namely, the sparsified adjacency matrix $A_{\textrm{sparsified}}$ satisfies $\textrm{E}[A_{\textrm{sparsified}}]=A$ with much less non-zero entries. In this way, we can greatly accelerate the training process since the speed bottleneck of GNNs is often the matrix operation on the large sparse matrix.
> > Specifically, the sparse-dense matrix multiplication (SpMM) is extremely slow due to the cache miss problem, which is caused by the irregular memory access pattern of SpMM [2]. The graph sparsification can greatly alleviate this problem since (1) it directly reduce the number of memory access (2) it reduces the FLOPs in SpMM operations.
> >
> > From our theoretical analysis,
> > we can introduce the "randomness" to both the input data (sampling-based graph sparsification) and activations (INT2 quantization), and the model performance should not be impacted a lot (both these two methods are unbiased compression methods with extra variances).
> > To verify this, we utilize the well-known effective-resistance based graph sparsification [1] to the ogbn-products along with the INT2 quantization to the activations.
> > The sparsified graph contains $66\%$ fewer edges compared to the original one.
> > The model here is a full-batch 3-layer 256-wide GraphSAGE (all results are averaged over ten random trials).
> > The results are as below:
> >
> > |  |  Accuracy (%) |  Training Throughput (epoch/s) | Data Mem (MB)     | Act Mem  (MB)     |
> > | :---------------------------------------: | :-----------------: | :---------------------------------: | :----------------: | :-----------------: |
> > | Baseline                                  | 78\.78±0.19         | 1\.047                              | 4811               | 16555               |
> > | INT2 Act                                  | 78\.79±0.12         |  0\.806 (23.0%↓)                    | 4811               | 1144 (14.5X)       |
> > |  INT2 Act+  <br>  Graph Sparsification    | 78\.87±0.10         |  1\.397 (33.4%↑)                    | 2583 (1\.9X)       | 1144 (14.5X)       |
> >
> > **We boost $33.4\%$ throughput and reduce both the input data memory and activation memory simultaneously, without any accuracy drop.**
> > **This impressive result is obtained directly following the insights of our theoretical analysis.**
> > Due to the time limit, currently, we only conduct the experiments on the ogbn-products since it is one of the largest datasets that can be examined on a single GPU with 24GB memory.
> > We will conduct similar experiments on more datasets and update the results once we finished them.
> > We will include these results in the final version.
> > We hope our discussion along with the new experimental results can address the reviewer's concern.
> >
> > [1] Spectral Sparsification of Graphs: Theory and Algorithms
> >
> > [2] GNNAdvisor: An Adaptive and Efficient Runtime System for GNN Acceleration on GPUs

---

> > ### Author Response · Authors · 2021-12-06
> > **Our dispute to the concern regarding the novelty 2**
> >
> > In our work, we claim that a 0.5% accuracy drop is acceptable in the introduction ("...EXACT can reduce the memory footprint of activations by up to 32× with roughly 0.5% loss in accuracy...").
> > In the initial review,
> > reviewer 2GtZ argues that a 0.5% accuracy drop from random projection is "significant". And thus, **reviewer 2GtZ implies that the random projection is not important and should not be accounted for the novelty of our work ("If we consider just quantization, then there is little novelty to this work I would argue").**
> >
> > **Our response**
> > 1. In the initial response, we first reiterate that a 0.5% accuracy drop is still acceptable without directly citing its source.
> >
> > 2. Reviewer 2GtZ suggests the term "acceptable" is hand-waving ("I think this is a stretch -- 0.5% is quite a lot to lose. How are we defining acceptable?")
> > To address the reviewer's concern, we then provide three detailed justification for supporting our claim. Specifically, we quote sentences from ActNN [1], BLPA [2], and OGB [3] to support our claim. **All of our quoted sentences directly point out that a 0.5% accuracy drop is even "negligible" in previous works.**
> >
> > 3. We also have quantitatively demonstrated the benefits of random projection for deeper GNNs (e.g., for GCNII on ogbn-arxiv, with the help of random projection, we further improve the maximal depth from 142 layers with only quantization to 273 layers; GCNII with both Random projection and INT2 quantization still outperforms all other baselines across different datasets).
> >
> > 4. Our second research contribution in the rebuttal is the theoretical analysis of why random projection and quantization have an interaction effect.
> > After the discussion, reviewer 2GtZ claims that although the interaction effect is interesting, the benefits of random projection are overstated relative to existing works.
> >
> > We would like to conclude that our stated benefits of random projections are (1) random projection has an interaction effect with quantization.
> > (2) "random projection is crucial for building deeper GNNs." (3) "it is worthwhile to trade such an accuracy degradation for reduced memory usage since, given a fixed memory budget, EXACT(RP+INT2) enables the usage of deeper GNN models on larger datasets".
> >
> > Here we also collect our provided evidence for supporting these three stated benefits.
> > Specifically, the first benefit is directly got supported by the quantitative results in Table 3 (a $0.5\%$ accuracy drop is still acceptable) and Figure 3 (the speed of EXACT is comparable to quantization only).
> > We support the second benefit by using the quantitative result that the random projection can further scale up the maximum depth of GCNII from 142 layers (quantization only) to 273 layers on the ogbn-arxiv.
> >  We support the third benefit by suggesting that "although there may be a larger accuracy drop (usually $\leq0.5\%$) after incorporating random projection, the accuracy drop can be compensated by the stronger expressiveness of deeper GNNs". Namely, from Table 3, GCNII with both random projection and INT2 quantization still outperforms all other baselines across different datasets.
> >
> > **To our knowledge, we have justified each of the claimed benefits of random projection with their respective experiments. If we have missed some details which lead to the statement "overstating the benefit of random projection relative to existing works", please let us know.**
> >
> > We respect the opinion that a 0.5% accuracy drop might be a problem in some important cases. However, according to previous works [1,2,3], we do not think a 0.5% accuracy drop is enough to discount the benefits of random projection. Hence, we disagree with the statement "consider just quantization, then there is little novelty to this work". With random projection in place, we argue the methodology of EXACT (random projection + quantization) is both novel and efficacious.
> >
> > [1]  ActNN: Reducing Training Memory Footprint via 2-Bit Activation Compressed Training
> >
> > [2] Backprop with Approximate Activations for Memory-efficient Network Training
> >
> > [3] Open Graph Benchmark: Datasets for Machine Learning on Graphs

---

> > ### Author Response · Authors · 2021-12-06
> > **Our dispute to the concern that our work does not add much to our knowledge as a community**
> >
> > Reviewer 2GtZ argues that the existing solutions (mainly quantization) can tackle the GNN's scalability issue well. However, we remark that
> > **no previous work has 1. extended the quantization methods to compress the activations of GNNs; and 2. studied the memory saving ratio, accuracy drop, and time overhead in actual experiments.
> > Given the importance of solving the scalability issue of GNNs and the strong results shown in this paper,
> > we think our work's value to the community is self-evident.**
> > **Moreover, many quantitative results reported in our paper are non-trivial to get and can be very insightful to the community.** For example, 1. how many bit-width is enough for activations of GNNs; 2. to what extent can we compress GNNs' activations; 3. what is the time overhead for training GNNs with compressed activations; 4. how many layers can be regarded as "moderately deep" such that the approximation error will not lead to a significant accuracy drop; 5. to what extent can we trim down the hardware requirement for training GNNs on large graphs.
> > We believe our work has provided useful data points and insights for future research.
> > **Thus, although we respect that Reviewer 2GtZ thinks the reported observation, namely, GNN is robust to extremely compressed activations, is not surprising, we disagree our work does not add much knowledge to the community.**
> >
> > We hope the reviewer can take the aforementioned points into account, including the two research contributions, practical contributions, additional experiments provided during the rebuttal, and the theoretical analysis.
> > We hope that our comments will change the opinion of the reviewer. Thanks for the time in providing feedback.

---

> > ### Comment · Reviewer_2GtZ · 2021-12-06
> > **Response to authors**
> >
> > > Reviewer 2GtZ argues that the existing solutions (mainly quantization) can tackle the GNN's scalability issue well
> >
> > This is completely untrue. I did not argue that quantization alone (e.g. quantization aware training) was a good solution to GNN scalability.
> >
> > > **we disagree our work does not add much knowledge to the community.**
> >
> > This is not something I argued about. The benefit to the community of these experiments is not something I am concerned about. However, I argue that the novelty from a research perspective is limited.
> >
> > > random projection is crucial for building deeper GNNs
> >
> > This is not a correct statement. Approaches such as those used by "[Training Graph Neural Networks with 1000 layers](https://ghli.org/publication/icml2021gnn1000/)" are already viable. My original argument about you overstating the contribution of your work relative to the existing literature remains true.

---

> > > ### Author Response · Authors · 2021-12-08
> > > **Reponse to the concern of Reviewer 2GtZ**
> > >
> > > >  (Author) Reviewer 2GtZ argues that the existing solutions (mainly quantization) can tackle the GNN's scalability issue well.
> > >
> > > > (Reviewer 2GtZ) This is completely untrue. I did not argue that quantization alone (e.g. quantization aware training) was a good solution to GNN scalability.
> > >
> > > After closer examination, we agree that reviewer 2GtZ did not argue that “quantization alone (e.g. quantization aware training) was a good solution to GNN scalability”.
> > > When we made our reply, we had the preconceived notion that our discussion was centered around lossy activation compression (e.g., quantization), which is not the situation. For this, we sincerely apologize.
> > >
> > > As a tangential comment, we do think quantization is a mainstream method for lossy activation compression solutions [1-6]. However, it is not excusable.
> > >
> > >
> > > > (Author) random projection is crucial for building deeper GNNs.
> > >
> > > > (Reviewer 2GtZ) This is not a correct statement. Approaches such as those used by "Training Graph Neural Networks with 1000 layers" are already viable. My original argument about you overstating the contribution of your work relative to the existing literature remains true.
> > >
> > >
> > > We agree that the approaches used in "Training Graph Neural Networks with 1000 layers" are very capable. However, we remark that our quote is originally from "As we mentioned in our response to Q2 (i.e., the quantitive results about the overall memory saving ratio), random projection is crucial for building deeper GNNs", **where we discuss about the role of random projection in the EXACT framework specifically**. Without random projection, for GCNII trained on the ogbn-arxiv dataset, the maximal depth without the OOM error decreases from 273 layers to 142 layers (int2 quantization only). Therefore, random projection is crucial for building deeper GNNs in our proposed framework. We did not intend it to describe building deeper GNNs in general.
> > >
> > > > (Author) we disagree our work does not add much knowledge to the community.
> > >
> > > > (Reviewer 2GtZ) This is not something I argued about. The benefit to the community of these experiments is not something I am concerned about. However, I argue that the novelty from a research perspective is limited.
> > >
> > > We would like to remark that the comment is quoted from one of our three responses to your comment "It (**novelty**) remains a big issue for me, and this work is **not adding much to our knowledge as a community** in my opinion".
> > >
> > > For the **concern about our work's novelty**, we provided our response in the posts entitled "Our dispute to the concern regarding the novelty 1" and "Our dispute to the concern regarding the novelty 2".
> > > As to the concern about our work's contribution to the knowledge as a community is also mentioned, that is why **we feel obliged to make our response** in the post entitled "Our dispute to the concern that our work does not add much to our knowledge as a community".
> > >
> > > ***
> > > To summarize this rebuttal, we have:
> > > 1. Provided the context in which we describe random projection to be "crucial" (i.e., specifically to EXACT)
> > > 2. Described why we feel obliged to mention our work's contribution to the "knowledge of the community" and pointed out where we explained the concern regarding the novelty of our work
> > >
> > > Most importantly,
> > >
> > > 3. We apologize for falsely concluding that the reviewer argued "existing solutions (mainly quantization) can tackle the GNN's scalability issue well".
> > >
> > > If there are any remaining concerns about the novelty of our work, please let us know. We are more than happy to provide additional information to help clarify the situation. We would be grateful if you tell us whether the response provided in "Our dispute to the concern regarding the novelty 1" and "Our dispute to the concern regarding the novelty 2" answer your concern about novelty and, if not, what we are lacking, so we can provide better clarification. Thank you for your time.
> > >
> > >
> > > [1]  Backprop with Approximate Activations for Memory-efficient Network Training
> > >
> > > [2] Gist: Efficient Data Encoding for Deep Neural Network Training
> > >
> > > [3] ActNN: Reducing Training Memory Footprint via 2-Bit Activation Compressed Training
> > >
> > > [4] Don’t waste your bits! squeeze activations and gradients for deep neural networks via tinyscript
> > >
> > > [5] Ultra-low precision 4-bit training of deep neural networks
> > >
> > > [6] Training and Inference with Integers in Deep Neural Networks

---

### Official Review · Reviewer_FTNh · 2021-11-04

**Correctness:** 4
**Technical Novelty And Significance:** 3
**Empirical Novelty And Significance:** 3
**Recommendation:** 8
**Confidence:** 3

**Main Review:**

Strengths:
1. The first work that explores the compressed activation in GNN training.
2. Provide comprehensive empirical evaluation and theoretical analysis of the trade-off among memory consumption, accuracy loss, and time overhead
3. Impressive results

Weaknesses:
There is no major weakness of this paper, but adding more comparisons against other techniques would be great.
1. Adding experiments to compare EXACT with other memory-saving techniques (e.g., gradient checkpointing, swapping)
2. Adding experiments to compare EXACT with sampling approaches given a fixed memory budget. Maybe you have to choose slightly different settings of different approaches due to their different memory requirements.


**Summary Of The Paper:**

This paper explores the training of GNN with compressed activation maps. It provides an optimized GPU implementation and comprehensively studies the trade-off among the memory saving, time overhead, and accuracy drop.
Experimental results show that the proposed framework can reduce the memory footprint of activations by up to 32x with only 0.2-0.5% accuracy drop and 10-25% time overhead.

**Summary Of The Review:**

This paper explores a novel direction and gets impressive results. I strongly recommend accepting this paper.

---

> ### Author Response · Authors · 2021-11-18
> **Response to Reviewer FTNh**
>
> **Q1. [comparisons against swapping and gradient checkpointing]**
>
> We add the comparison experiment against the swapping and gradient checkpointing in the updated version (Figure 3 and Appendix I.2). Both swapping and gradient checkpointing are lossless methods, so they do not have any accuracy drop. Below we summarize their trade-off among the space and speed.
>
> Swapping: For swapping, we offload all activation maps to CPU memory. Thus, it has the highest compression ratio as long as the CPU memory is enough. However, as shown in Figure 3, its main drawback is that its extra time overhead can go up to 80%, which is not feasible in practice. This is because moving data between CPU and GPU is much expensive than directly quantizing tensors on GPU.
>
> Gradient checkpointing: For implementation convenience, we utilize torch.utils.checkpoint.checkpoint to insert the checkpoint at each GNN layer. Its time overhead is comparable to EXACT, however, the memory compression ratio of activations is 1.7∼2.3× (Table 20), which is significantly smaller compared to EXACT.
>
> In summary, swapping and gradient checkpointing do not have any accuracy drop. However, compared to EXACT, swapping is too slow, and the compression ratio of gradient checkpointing is not large enough.
>
> **Q2. [Compare EXACT with sampling approaches given a fixed memory budget]**
>
> We apologize that we do not add the experiments since directly comparing these two methods are unfair due to the following reason.
> EXACT can only compress the activations. In contrast, sampling-based methods can simultaneously reduce the memory of the input data, activations, and activation gradients since they directly reduce the number of node embeddings retained in the memory.
> However, as shown in Table 3, one main drawback is that the accurancy drop of sampling-based method may be greater than EXACT.
> As we shown in the paper, EXACT and sampling-based methods can be applied over each other. Moreover, we can utilize EXACT to enlarge the maximal batch size of sampling based method.
>
> Thus, instead of directly comparing EXACT with sampling based methods, we present a case study of scaling up the batch size of GraphSAINT with EXACT on the ogbn-products dataset in Appendix I.5 (the memory budget is fixed). As shown in Table 2, EXACT may even improve the test accuracy with 3× larger batch size (79.03%->79.16%).
> EXACT can actually scale up the batch size to more than 3× larger. However, we found that if we further scale up the batch size to 4×, there is an accuracy drop compared to the baseline with the original batch size, regardless of applying EXACT or not. We note that for a fair comparison, we do not change any hyperparameters. However, this might be problematic since when the batch size is large, similar to other neural networks, the learning rate also needs to be tuned [1, 2]. Finding the optimal learning rate under different batch sizes requires lots of computational resources and human efforts, which is beyond the scope of this paper. We leave it as future work.
>
> ============================update========================
>
>
> We compare full batch GraphSAGE with EXACT to subgraph sampling methods under a fixed activation memory budget (**Appendix I.5.1**). Detailed experimental settings can be found in **Appendix I.5.1**.
> For Reddit and Flickr, full batch training with EXACT can outperform subgraph sampling methods.
> For Yelp, subgraph sampling methods are  better than full batch training with EXACT.
> However, we note that this gap is not from EXACT.
> As shown in Table 3, without EXACT, the F1-micro of full batch GraphSAGE is $62.05\%$, which is much lower than that of subgraph sampling methods.
> In summary, subgraph sampling methods may outperform the full batch training on some datasets (e.g., Yelp).
> In this case, we think full batch training with EXACT cannot outperform subgraph sampling methods.
> If this is not the case, as shown in following tables, full batch training with EXACT can outperform subgraph sampling methods under a fixed activation memory usage.
>
>
> [1] Imagenet training in minutes
>
> [2] Large batch optimization for deep learning: Training bert in 76 minutes

---

### Author Response · Authors · 2021-11-20
**Summarization of the revision**

We thank all reviewers for the constructive reviews.
We have revised the paper accordingly and marked the modifications in blue color for visibility. The changes are as follows.

1. We add the comparision experiments against swapping the gradient checkpointing (Figure 3 and Appendix I.2).

2. We add a case study of scaling up the batch size of GraphSAINT with EXACT on the ogbn-products dataset (Appendix I.5).

3. We report the memory consumption of input data, activations, the gradients, respectively. We also report the overall memory saving ratio of EXACT (Appendix H.2 Table 18 and Table 19, Appendix H.2)

4. We fix some small typos, e.g., throughout -> throuput in Figure 3, and the typos pointed out by Reviewer eNSi. We also add the reference to one Neurips 2021 paper AC-GC.

==================update======================

5. We compare full batch GraphSAGE with EXACT to GraphSAINT and Cluster-GCN in a standalone way (Appendix I.5.1)

6. We add a case study for showing that the activations of GNNs can be aggressively compressed even under a much smaller batch size (Appendix E.4)

7. We update the related work and discuss the potential benefit of JPEG-ACT, CuSZ, and AC-GC for GNNs (Appendix B).

=============================================

We apologize that we have not released our code in the initial submission. Our code is available at https://anonymous.4open.science/r/exact-revd.

If there is any further question, please let us know.

---

### Decision · Program_Chairs · 2022-01-20

**Decision:**

Accept (Poster)

**Comment:**

There are numerous known methods for memory reduction used in CNNs.
This paper takes two such---quantization (Q) and random projection (RP)---and applies them to GNNs.  This is a novelty, but I agree with the reviewers: on its own this novelty would not be "surprising" enough to report at ICLR.

The paper further goes to show empirically that these methods, when applied to a reasonable set of datasets, do indeed produce their predicted memory reductions (unsurprising) with a small ($\approx 0.5\%$) drop in accuracy (surprising, in the sense of not being something one could predict without doing the experiment).

All of the above is in one sense "just engineering", with only a small inventive step.  Any real-world deployment of GNNs would, if an army of engineers were available, naturally implement quantization and RP in order to see what kind of improvements they might make.  This would be just two more hyperparameters (R,B) to add to the sweep, and the deployment would vary them until the required accuracy was achieved in the minimum time (OOM is a red herring - one would vary batch size, other compression, or ship values to CPU in order to make progress).

However, "simply adding two more hyperparameters" is a significant increase in the deployment burden, which is where the paper's third contribution comes in: the theoretical analysis of the effects of the two processes, with straightforward but nonobvious calculations of the effect on gradient variance of the two processes, and, usefully, their interaction.  The value of this theory is twofold: first, it gives us new tools to analyse such processes; and second, it allows us to be much more judicious in the selection of these hyperparameters.

In all, the reviewers' objection of no great novelty in porting ideas from CNN  to GNN is sustained; but the authors' claim as to the value of the theory is sustained, and no reviewer provides prior art to dispute the novelty of the theory calculations.

The revised paper has already expanded the key sections in Appendix E, and added welcome experiments which strengthen the paper.  I would encourage a final copy (and certainly the poster presentation) to emphasize some of the insights over the raw experimental numbers.  As the authors hint, those numbers are subject to vagaries of what PyTorch happens to implement, while the underlying analyses are a little longer lasting.

Some other comments:

A lot of discussion time was spent on the question of whether 0.5% is negligible.  This is entirely application dependent, and is part of the hyperparameter/architecture tuning process.

  - the extra time overhead of swapping "can go up to 80%, which is not feasible in practice".
  Not so: if choosing between OOM or 1.8x slowdown, I will of course choose the latter.
  - "for a fair comparison, we do not change any hyperparameters"
  Again, not relevant: in a real application (which is where this paper contributes), we of course change the learning rate when batch size changes.
  - "the accuracy drop of sampling may be greater than EXACT"
  Again, whether that drop is too much depends entirely on the actual application.

And please do take a look at typos/grammar/English etc.